# Defence-mediated phloem restriction of a plant virus facilitates insect transmission

Yuzhen Mei[1,2], Yaqin Wang[2], Fangfang Li[1], Rosa Lozano-Durán ®[3] ✉ & Xueping Zhou ®[1,2] ✉

Plant viruses are causal agents of devastating diseases in crops and pose a threat to food security. Viruses transmitted by phloem-feeding insects are frequently restricted to the phloem; the mechanism determining this tissue tropism and its consequences for the viral cycle have remained elusive. Here we show that phloem restriction of tobacco curly shoot virus (TbCSV) depends on the allele of the *C4* gene it carries. The Y35 allele produces a plasma membrane-associated C4 protein; TbCSV(Y35) is phloem-limited. The Y41 allele, however, produces two C4 variants, targeted to the plasma membrane and chloroplasts, respectively. Chloroplast-localized C4 interferes with salicylic acid-mediated defenses, which causes a destabilization of PENETRATION3 (PEN3), a subsequent decrease in callose deposition, and the escape of TbCSV(Y41) to the surrounding parenchyma cells. Interestingly, Y41 expands the host range of TbCSV, but Y35 is prevalent in nature. We determine that phloem restriction favors acquisition and transmission by the insect vector, conferring a competitive advantage to TbCSV(Y35). Importantly, PEN3 also determines phloem restriction of an RNA virus. In summary, we demonstrate that PEN3 activity and likely callose deposition confine viruses to the phloem, which favors viral spread by facilitating acquisition by the insect vector.

Plant viruses cause devastating diseases in crops worldwide, posing a serious threat to global food security. Most plant viruses are transmitted by phloem-feeding insects; following transmission by the vector, many viruses remain restricted to the phloem[1–4]. Despite the prevalence of this tissue tropism, the underlying molecular mechanisms as well as its significance for viral performance have so far remained elusive.

Insect-transmitted viruses from the *Geminiviridae* family are causal agents of economically relevant diseases affecting cash and staple crops in tropical and temperate regions of the world. Most geminiviruses are transmitted by the phloem-feeding whitefly *Bemisia tabaci*, and most of these are phloem-limited. Many species within this family encode a protein named C4, which is versatile and essential for full infectivity, suppressing plant anti-viral defenses and mediating

symptom development[5–7]. C4 proteins from different geminiviruses possess different combinations of targeting signals and show specific subcellular localizations[6]; the C4 protein encoded by tomato yellow leaf curl virus (TYLCV), for example, displays a dual localization at the plasma membrane and in chloroplasts, moving from the former to the latter during infection[8]. At the plasma membrane, C4 from TYLCV suppresses the cell-to-cell spread of silencing, while in chloroplasts it interferes with the activation of anti-viral salicylic acid (SA)-dependent defenses[8,9]. This subcellular compartmentalization that enables multi-functionality of C4 seems to be relevant for the infection, as geminiviruses have evolved independent strategies to accomplish it. Recently, it has been demonstrated that some geminiviruses encode two variants of the C4 protein in their *C4* gene, differing in the two N-terminal residues only, resulting from translation initiation from two

[1]State Key Laboratory for Biology of Plant Diseases and Insect Pests, Institute of Plant Protection, Chinese Academy of Agricultural Sciences, Beijing, China. [2]State Key Laboratory of Rice Biology and Breeding, Institute of Biotechnology, Zhejiang University, Hangzhou, Zhejiang, China. [3]Department of Plant Biochemistry, Centre for Plant Molecular Biology (ZMBP), Eberhard Karls University, D-72076, Tübingen, Germany. ✉e-mail: rosa.lozano-duran@uni-tuebingen.de; zzhou@zju.edu.cn

alternative in-frame start codons; one of these variants localizes to the plasma membrane, while the other is translocated to chloroplasts[10]. This is the case of tomato golden mosaic virus (TGMV), whose plasma membrane-localized C4 protein inhibits receptor kinases-mediated anti-viral defenses and triggers leaf curling symptoms, while the chloroplast-localized variant promotes degradation of this organelle and hence yellowing symptoms[10].

An interesting case is that of tobacco curly shoot virus (TbCSV), of which different isolates coexisting in the field differ in their capacity to encode either one or two C4 variants[10], offering a unique opportunity to evaluate the biological relevance of this diverging property. We show that phloem restriction of TbCSV depends on the allele of the *C4* gene it carries. The Y35 allele (TbCSV(Y35)) produced a plasma membrane-associated C4 protein that is phloem-limited. The Y41 allele TbCSV(Y41) produced two C4 variants, targeting the plasma membrane and chloroplasts, respectively, that compromise phloem restriction. The chloroplast-localized C4 causes a destabilization of PEN3 (an ATP-binding cassette transporter), a subsequent decrease in callose deposition, and the escape of TbCSV(Y41) to the surrounding parenchyma cells. Importantly, we demonstrate that PEN3 activity, likely through callose deposition, confines viruses to the phloem, which favors viral spread by facilitating acquisition by the insect vector.

## Results

### TbCSV (Y41) isolates have a lower epidemic rate

Most TbCSV isolates encoding only one C4 variant (localized to the plasma membrane) infect plant species from the *Solanaceae* family exclusively, while those isolates encoding two variants can additionally infect members of the *Amaranthaceae*, *Cucurbitacea*, and *Compositae* families (Supplementary Fig. 1, Supplementary Table 1). Elegant allele-swap experiments have demonstrated that the presence of the second, chloroplast-localized C4 variant is sufficient to expand the host range of the virus[10]. Two TbCSV isolates found co-existing in the same geographical area (Yunnan, China) were selected for further comparative characterization: TbCSV(Y35), which carries the Y35 C4 allele, producing only one, plasma membrane-associated, C4 protein, and TbCSV(Y41), which carries the Y41 C4 allele, producing two C4 proteins differing in the two N-terminal amino acids and targeted to the chloroplast and the plasma membrane, respectively.

Intriguingly, despite the increase in potential hosts that the Y41 allele enables, isolates carrying the Y35 allele are prevalent in China (Fig. 1a, Supplementary Table 2), pointing to a potential trade-off imposed by the Y41 allele. To dissect the impact of having two C4 variants for viral performance and identify potential negative effects, a TbCSV(Y35) virus carrying the Y41 C4 allele (TbCSV(Y35-2C4); Fig. 1b) was generated. This recombinant virus triggered the typical yellowing symptoms associated with the chloroplast-localized C4 protein in infected tomato plants (Fig. 1c). All three viruses accumulated to similar levels, indicating that viral performance in the plant is not differentially affected by these C4 alleles (Fig. 1d).

In the absence of changes in the infection per se, effects on insect transmission can account for the differential prevalence of the isolates in nature. To test whether transmissibility was affected by the presence of the two C4 variants, whiteflies were put to feed on tomato plants infected with each of these three viruses, and the presence of the pathogen was detected in the midgut and salivary glands of the insects (Fig. 1e–h). These experiments clearly showed that TbCSV isolates expressing two C4 variants (TbCSV (Y41), TbCSV (Y35-2C4)) accumulated to a lower extent in the insect vector; of note, this is not due to differences in the viral capsid that might affect acquisition, since accumulation of virions added to the insect feed in the whitefly organs was similar (Fig. 1i, Supplementary Fig. 2). As expected from the decreased presence of the virus in the insect vector, transmission

efficiency of the isolates with two C4 variants was diminished significantly (Fig. 1j, Supplementary Fig. 3).

### The Y41 allele compromises phloem restriction

Chloroplast-localized C4 has been linked to yellowing symptoms in TGMV[10]. Intriguingly, the presence of the Y41 or the Y35-2C4 alleles in TbCSV resulted in yellowing symptoms in the leaf lamina, not only in tomato (Fig. 1c), but also in the experimental host *Nicotiana benthamiana* (Fig. 2a). This phenotype prompted the idea that the chloroplast-localized C4 might be acting outside the phloem, the tissue TbCSV is considered restricted to. Differences in symptoms do not rely on changes in viral accumulation in *N. benthamiana* either (Fig. 2b). Strikingly, immunolocalization of the viral capsid protein (CP) in cross-sections of leaves (Fig. 2c) or leaf veins (Fig. 2d, e) from infected *N. benthamiana* plants demonstrates that viruses encoding a chloroplast-localized C4 protein (TbCSV(Y41), TbCSV(Y35-2C4)) are not only detected in the phloem of both sides of the bicollateral vascular bundles but also in parenchyma cells outside of the phloem in over 70% of the images, as opposed to TbCSV(Y35) detected only in the phloem, which carries the Y35 allele. In particular, the comparison between TbCSV(Y35) and TbCSV(Y35-2C4) allows the conclusion that the chloroplast-localized C4 is sufficient to compromise phloem restriction, enabling the virus to reach parenchyma cells.

### Phloem restriction likely requires PEN3-dependent callose deposition

The yellowing of the leaf lamina observed in plants infected with TbCSV carrying the Y41 allele is reminiscent of that displayed by *pen3-1* mutant *Arabidopsis thaliana* plants upon inoculation with the powdery mildew *Erysiphe cichoracearum*[11]. *PEN3* encodes a plasma membrane-localized ATP-binding cassette transporter required for pathogen-induced callose deposition, which impairs pathogen penetration in a salicylic acid (SA)-dependent manner[11,12]. The PEN3 orthologue in *N. benthamiana*, NbPEN3, also localizes to the plasma membrane (Fig. 3a). Since callose deposition at plasmodesmata is known to restrict movement of viruses[4,13–17], we sought to investigate whether PEN3 is involved in the phloem limitation of TbCSV(Y35). For this purpose, two independent *NbPEN3* knock-out *N. benthamiana* lines were obtained through CRISPR-Cas9-mediated genome editing (Fig. 3b, Supplementary Fig. 4); these mutant lines showed largely decreased levels of callose (Supplementary Fig. 4b, c). Remarkably, *nbpen3* mutants developed yellowing in the leaf lamina during infection with all three TbCSV viruses, regardless of their capacity to encode a chloroplast-localized C4 protein; viral accumulation, nevertheless, remained unchanged (Fig. 3c, d). In agreement with the correlation between phloem escape and leaf yellowing previously established, all three viruses could exit the phloem and reach parenchyma cells in *nbpen3* mutant lines (Fig. 3e). Phloem restriction of TbCSV(Y35) in wild-type plants was associated with strong virus-induced PEN3-dependent callose deposition surrounding the vasculature and higher virus accumulation in the phloem (Fig. 3f, Supplementary Fig. 5), which was not observable in wild-type plants infected with TbCSV(Y41) nor in *nbpen3* mutants in any case. Our results, therefore, indicate that PEN3-mediated callose deposition is triggered by the TbCSV(Y35) infection and can effectively restrict this viral strain and increase viral titer in the phloem, while TbCSV(Y41) can likely avoid or suppress this response through the action of its chloroplast-localized C4 protein. To test whether callose deposition limits virus to the phloem, we treated TbCSV(Y35)-infected *N. benthamiana* plants with 2-deoxy-D-glucose (2-DDG), an inhibitor of callose synthesis. Immunoblot and confocal results showed that TbCSV(Y35) escapes the phloem to the surrounding parenchyma cells in 2-DDG-treated leaves, which correlates with lower callose accumulation (Supplementary Fig. 6). These results indicate a direct link between phloem limitation and callose deposition

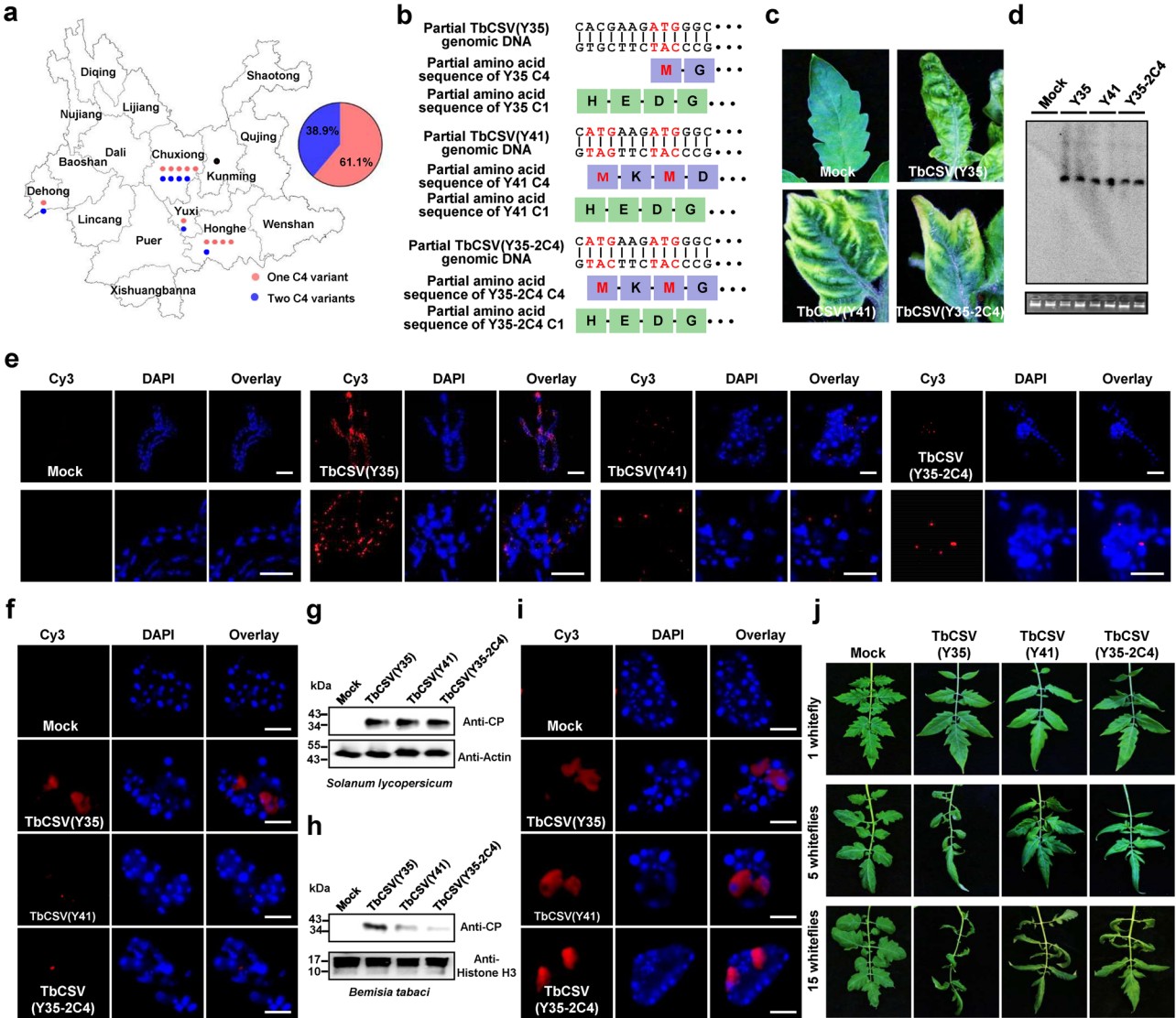

**Fig. 1 | TbCSV isolates expressing two C4 variants have a lower epidemic rate.**
**a** TbCSV isolates containing one C4 variant only is prevalent in Yunnan, China. The proportion of TbCSV isolates containing two C4 variants is indicated in blue; the proportion of TbCSV isolates harboring only one C4 variant is indicated in red. TbCSV isolates shown in Fig. 1a were collected from tomato plants in Yunnan, China, between 2010 and 2017. **b** Nucleotide sequences complementary to the virion-sense at the start of the *C4* gene of TbCSV(Y35), TbCSV(Y41), and TbCSV(Y35-2C4). The encoded amino acid sequences of C4 and C1, encoded in the overlapping gene, are shown. The ATG of the *C4* gene is indicated in red letters. **c** Symptoms caused by TbCSV(Y35), TbCSV(Y41), and TbCSV(Y35-2C4) in tomato plants at 28 days post-inoculation (dpi). **d** Southern blot analysis of TbCSV(Y35), TbCSV(Y41), and TbCSV(Y35-2C4) accumulation in tomato plants at 28 dpi. Total DNA was hybridized with a *TbCSV CP* probe. Total nucleic acids (50 µg DNA) were extracted from tomato plants inoculated with TbCSV(Y35), TbCSV(Y41), or TbCSV(Y35-2C4) infectious clones at 28 dpi. Total genomic DNA visualized by ethidium bromide staining is shown as a loading control. **e** Immunofluorescence detection of TbCSV in midguts of Zhejiang II whiteflies after feeding on TbCSV(Y35)-, TbCSV(Y41)-, or TbCSV(Y35-2C4)-infected tomato plants. TbCSV was detected using a monoclonal antibody against the TbCSV coat protein (CP), followed by a commercial 549-conjugated secondary antibody (red); nuclei are stained with DAPI (blue). At least 10 midguts were examined for each treatment. Representative images are shown. Scale bar = 100 µm. **f** Immunofluorescence

detection of TbCSV in primary salivary glands of Zhejiang II whiteflies after feeding on TbCSV(Y35)-, TbCSV(Y41)-, or TbCSV(Y35-2C4)-infected tomato plants. As in (**e**). At least 10 primary salivary glands were examined for each treatment. Representative images are shown. Scale bar = 20 µm. **g**, **h** Immunoblot analysis of virus accumulation in whiteflies after feeding on TbCSV(Y35)-, TbCSV(Y41)-, or TbCSV(Y35-2C4)-infected tomato plants. **g** Accumulation of TbCSV(Y35), TbCSV(Y41), and TbCSV(Y35-2C4) CP in virus-infected tomato plants. Actin was used as a loading control. **h** TbCSV(Y35), TbCSV(Y41), and TbCSV(Y35-2C4) CP accumulation in whiteflies after feeding on virus-infected tomato plants. Histone H3 was used as a loading control. **i** Immunofluorescence detection of TbCSV in primary salivary glands of Zhejiang II whiteflies after feeding on artificial feed containing TbCSV(Y35), TbCSV(Y41), or TbCSV(Y35-2C4) virions. At least 10 primary salivary glands were examined for each treatment. Representative images are shown. Scale bar = 20 µm. **j** Transmission efficiency of TbCSV(Y35), TbCSV(Y41), or TbCSV(Y35-2C4) by Zhejiang II whiteflies. The upper, middle, and lower panels indicate different numbers of viruliferous whitefly adults (1, 5, or 15, respectively) used per plant. Tomato plants were given an inoculation access period of 72 h with 1, 5, or 15 viruliferous whiteflies using clip cages in three independent biological replicates. Non-viruliferous whiteflies were used as a mock treatment. Photographs were taken 14 days after the inoculation access period. Experiments in (**d**−**h**, **i**) were repeated three times with similar results.

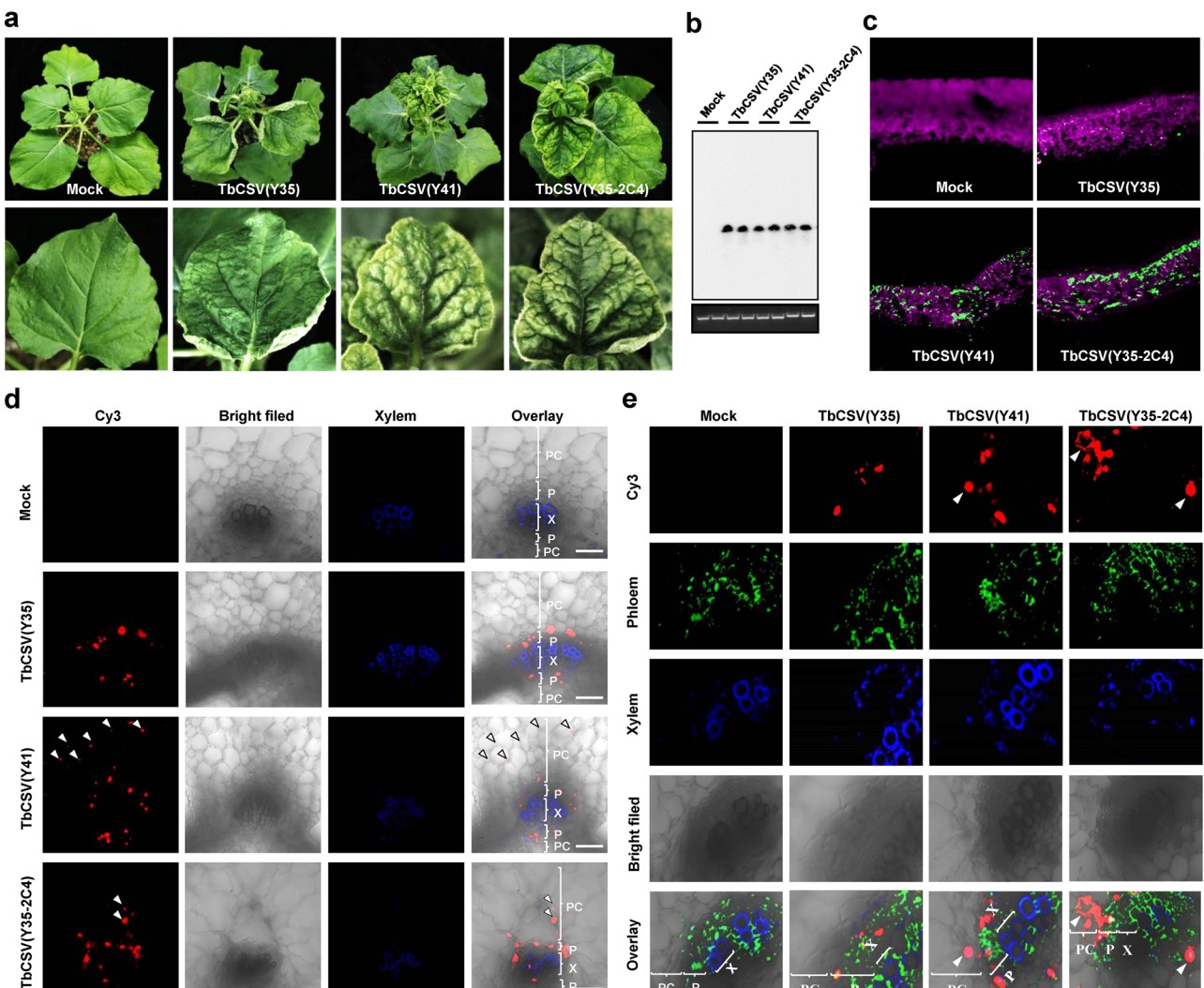

**Fig. 2 | The ability of TbCSV to express a chloroplast-localized C4 compromises phloem restriction. a** Viral symptoms induced by TbCSV(Y35), TbCSV(Y41), and TbCSV(Y35-2C4) in *Nicotiana benthamiana* plants at 14 days post-inoculation (dpi). **b** Southern blot analysis of TbCSV(Y35), TbCSV(Y41), and TbCSV(Y35-2C4) accumulation in *N. benthamiana* plants at 14 dpi. Total DNA was hybridized with a *TbCSV CP* probe. Total nucleic acids (50 μg DNA) were extracted from *N. benthamiana* plants inoculated with TbCSV(Y35), TbCSV(Y41), or TbCSV(Y35-2C4) infectious clones at 14 dpi. Total genomic DNA visualized by ethidium bromide staining is shown as a loading control. **c** Immunofluorescence detection of TbCSV in leaves of *N. benthamiana* plants infected by TbCSV(Y35), TbCSV(Y41), or TbCSV(Y35-2C4). The distribution of TbCSV(Y35), TbCSV(Y41), or TbCSV(Y35-2C4) was visualized in a cross-section using an antibody against the TbCSV coat protein (CP) (green). **d** TbCSV strains expressing two C4 variants (Y41 and Y35-2C4) overcome phloem restriction and reach parenchyma tissue in *N. benthamiana* plants. The distribution of TbCSV(Y35), TbCSV(Y41), or TbCSV(Y35-2C4) was visualized in a cross-section using an antibody against TbCSV CP (red). Autofluorescence of highly lignified

tissues is shown in blue. PC, parenchyma; P, phloem; X, xylem. White arrowheads indicate TbCSV in parenchyma cells. Scale bar = 50 μm. Statistical analysis showed that TbCSV(Y35) is strictly limited to the phloem in *N. benthamiana* plants, but TbCSV(Y41) and TbCSV(Y35-2C4) can be observed overcoming phloem restriction and reaching parenchyma cells in over 70% of the images. At least eight individual samples of TbCSV(Y35)-infected *N. benthamiana* plants, five individual samples of TbCSV(Y41)-infected *N. benthamiana* plants and six individual samples of TbCSV(Y35-2C4)-infected *N. benthamiana* plants from three independent biological replicates were used for immunochemistry and statistical analysis. **e** Immunofluorescence detection of TbCSV in leaf vein sections of *AtSWEET11::GFP* transgenic *N. benthamiana* plants. The distribution of TbCSV(Y35), TbCSV(Y41), or TbCSV(Y35-2C4) and phloem cells was visualized using antibodies against TbCSV CP (red) or GFP (green). Autofluorescence of highly lignified tissues is shown in blue. PC, parenchyma; P, phloem; X, xylem. White arrowheads indicate TbCSV in parenchyma cells. Scale bar = 50 μm. Experiments in (**b**–**d**, **e**) were repeated at least three times with similar results.

and suggest that the latter plays a critical role in phloem restriction of viruses.

## A Y41-encoded C4 protein variant enables phloem escape

With the aim to elucidate how TbCSV(Y41) overcomes the PEN3-dependent phloem restriction, we first measured *NbPEN3* transcript and protein accumulation in the presence of TbCSV(Y35), TbCSV(Y41), or TbCSV(Y35-2C4). While no changes were observed in the levels of *NbPEN3* mRNA (Fig. 4a), the NbPEN3

protein accumulation was largely decreased in the presence of TbCSV encoding the chloroplast-localized C4 (Fig. 4b). This effect could also be observed at the tissue level: while TbCSV(Y35) and PEN3 co-localized, cells infected with TbCSV(Y41) or TbCSV(Y35-2C4) were devoid of detectable PEN3 (Fig. 4c). Chloroplast-localized C4 has no effects on the *NbPEN3* transcription, but was sufficient to trigger the reduction in NbPEN3 content and callose deposition, which could also be observed in transgenic *N. benthamiana* plants expressing the C4 proteins encoded by the Y41

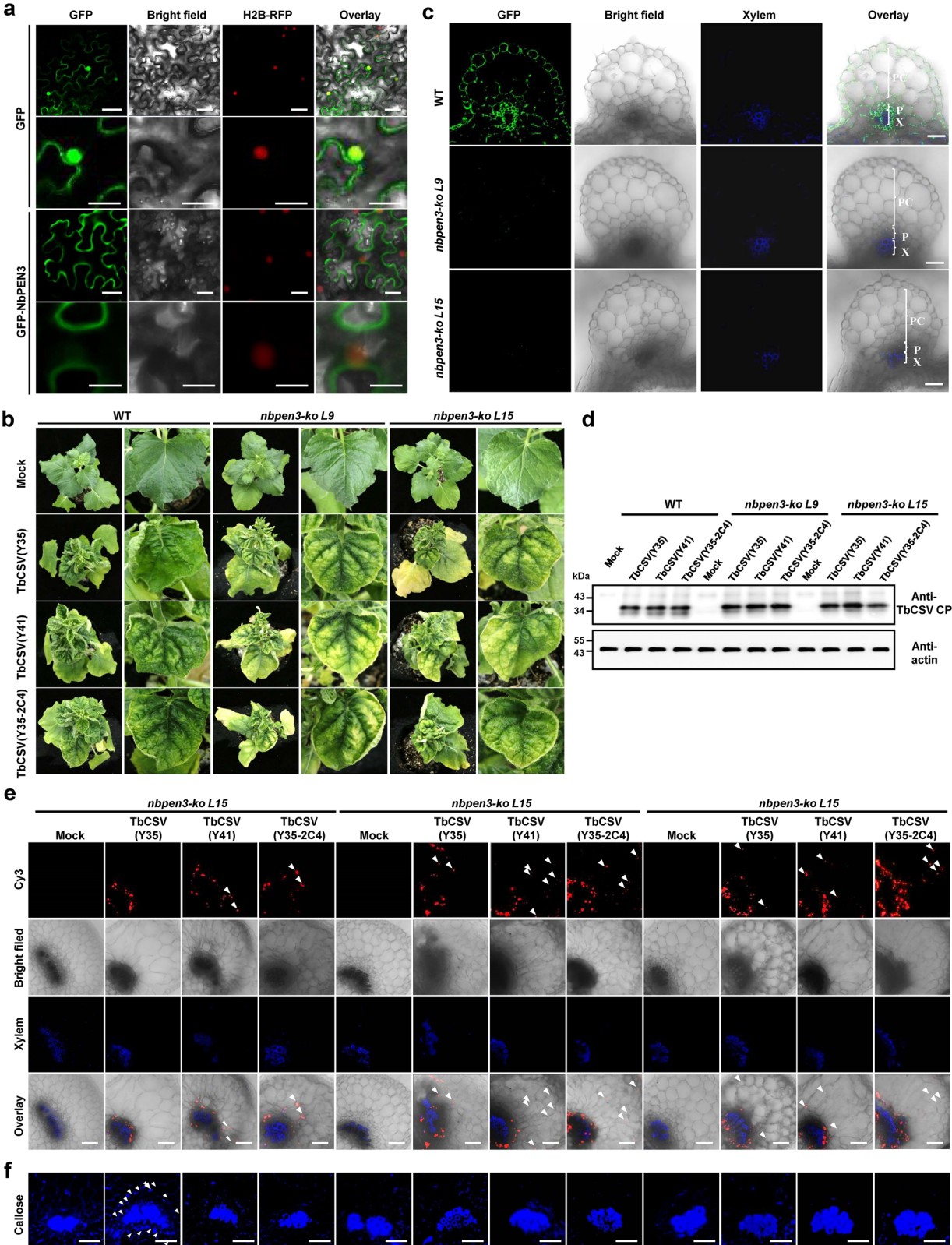

or the Y35-2C4 alleles, but not the Y35 allele (Fig. 4d, e, Supplementary Fig. 7).

Chloroplast-dependent SA-mediated defenses have been shown to play an anti-geminiviral role, and to be suppressed by C4 orthologues from other geminivirus species[8,18]. In order to test whether the TbCSV-encoded chloroplast-localized C4 interferes with the activation of SA signaling, expression of the SA-responsive gene *NbPR1* was evaluated in the transgenic plants expressing the different C4 proteins. As shown in Fig. 4f, the presence of a chloroplast-localized C4 version is sufficient to hamper the accumulation of the *NbPR1* transcript, hence suggesting that this viral protein can suppress SA defenses. To elucidate the underlying molecular mechanism, we tested the interaction between the C4 protein encoded by the Y41 allele and NbCAS, a proven target of other geminivirus-encoded chloroplast-localized C4

**Fig. 3 | NbPEN3-dependent callose deposition determines phloem restriction of TbCSV. a** Cellular distribution of NbPEN3-GFP in epidermal cells of H2B-RFP transgenic *N. benthamiana* plants. Scale bar = 50 µm. **b** Immunofluorescence detection of NbPEN3 in leaf vein sections of wild-type (WT) and *nbpen3* knock-out *N. benthamiana* plants. The distribution of NbPEN3 was visualized in a cross-section using an antibody against PEN3 (green). Autofluorescence of highly lignified tissues is shown in blue. PC, parenchyma; P, phloem; X, xylem. Scale bar = 50 µm. **c** Viral symptoms induced by TbCSV(Y35), TbCSV(Y41), and TbCSV(Y35-2C4) in WT and *nbpen3* knock-out *N. benthamiana* plants at 14 days post-inoculation (dpi). **d** Immunoblot analysis of TbCSV(Y35), TbCSV(Y41), and TbCSV(Y35-2C4) coat protein (CP) accumulation in WT and *nbpen3* knock-out *N. benthamiana* plants at 14 dpi. TbCSV CP was detected using a monoclonal antibody. Actin was used as a loading control. **e** Immunofluorescence detection of TbCSV in leaf vein sections of WT and *nbpen3* knock-out *N. benthamiana* plants. The distribution of TbCSV(Y35), TbCSV(Y41), or TbCSV(Y35-2C4) was visualized in a cross-section using an antibody against the TbCSV CP (red). Autofluorescence of highly lignified tissues is shown in blue. White arrowheads indicate TbCSV in parenchyma cells. Scale bar = 50 µm. In TbCSV(Y35)-infected *pen3-ko N. benthamiana* plants, over 50% of images show virus signal in parenchyma cells. At least eleven individual samples of TbCSV(Y35)-infected *pen3-ko N. benthamiana* plants, ten individual samples of TbCSV(Y41)-infected *pen3-ko N. benthamiana* plants and twelve individual samples of TbCSV(Y35-2C4)-infected *pen3-ko N. benthamiana* plants from three independent biological replicates were used for immunochemistry and statistical analysis. **f** Callose deposition in leaf vein sections of WT and *nbpen3* knock-out *N. benthamiana* plants infected by TbCSV(Y35), TbCSV(Y41), or TbCSV(Y35-2C4) at 14 dpi. Callose was stained by using aniline blue. White arrowheads indicate virus-induced callose deposition around the vasculature. Scale bar = 50 µm. Experiments in (**a, c–f**) were repeated at least three times with similar results.

proteins[8]. As shown in Fig. 4g, h, Y41 C4 interacts with NbCAS. In addition, other C4 proteins have been shown to compromise chloroplast integrity[10], we observed the chloroplast ultrastructure of wild-type and *35S::TbCSV(Y35) C4* transgenic *N. benthamiana* plants is not different from that of transgenic *N. benthamiana* plants expressing the chloroplast-localized C4, but fewer osmiophilic granules can be detected following osmic acid staining in *35S::TbCSV(Y41) C4* and *35S::TbCSV(Y35-2C4) C4* transgenic plants (Supplementary Fig. 8). These results suggest that the chloroplast-localized C4 might interfere with SA signaling through the physical interaction with NbCAS and/or by disrupting the redox balance in chloroplasts. Importantly, the stability of NbPEN3 is regulated in a SA-dependent manner, since NbPEN3 was destabilized in the absence of SA in transgenic *NahG* plants, encoding a bacterial salicylate hydroxylase that degrades this plant hormone (Fig. 4i, j, Supplementary Fig. 9a). Consistent with these results, SA treatment compromised the degradation of NbPEN3, which confirms the critical role of SA in the regulation of NbPEN3 stability (Supplementary Fig. 9b). As previously observed for the *nbpen3* mutant lines, *NahG* plants also displayed yellowing of the leaf lamina when infected with TbCSV(Y35), which correlated with the escape of this virus outside of the phloem and the absence of callose deposition in infected tissues (Fig. 4k, l, Supplementary Fig. 10). Taken together, these results indicate that the chloroplast-localized C4 protein encoded by the Y41 allele reduces the PEN3 content in infected cells, most likely through the suppression of SA signaling, since SA depletion is sufficient to abrogate PEN3 accumulation and complement the lack of a chloroplast-localized C4 in overcoming phloem restriction.

### PEN3-facilitated phloem restriction favors acquisition of the virus by the insect vector

The Y41 C4 allele has a negative impact on insect transmission of the virus (Fig. 1e–j); since the chloroplast-localized C4 protein variant encoded by the Y41 allele suppresses SA signaling, leads to decreased PEN3 accumulation, and in turn enables viral phloem escape, and considering that the insect vector, *B. tabaci*, is a phloem feeder, it could be hypothesized that invasion of parenchyma would lead to viral dilution, which may ultimately diminish transmission efficiency. To test this idea, we first knocked out *PEN3* in tomato (*SlPEN3*) (Supplementary Fig. 11) and evaluated the effect of this mutation on the viral infection. As shown in Fig. 5a, *slpen3* plants displayed yellowing of the leaf lamina upon TbCSV infection irrespective of the C4 allele carried by the virus, although total viral accumulation was not affected, as observed for the *nbpen3* mutants (Supplementary Fig. 12). Also in this case, the knock-out of *SlPEN3* enabled TbCSV(Y35) phloem escape (Fig. 5b). Importantly, the expansion of tissue tropism facilitated by *SlPEN3* knock-out had a negative impact on insect acquisition, as demonstrated by the lower viral accumulation in the midgut of whiteflies fed on these plants (Fig. 5c, d, Supplementary Fig. 11). We next performed the transmission assays with wild-type and *SlPEN3*

knock-out plants, consistent with the decreased accumulation of the virus detected in the insect vector fed on *slpen3* tomato plants, transmission efficiency from *slpen3* tomato plants was diminished for all isolates (Fig. 5e–g). These results suggest that PEN3-mediated phloem restriction promotes viral spread by favoring the acquisition of the virus by the insect vector.

### PEN3 is a general determinant of viral phloem restriction

To test whether PEN3 mediates phloem restriction in other geminiviruses, we inoculated WT and *nbpen3 N. benthamiana* plants with tomato yellow leaf curl virus (TYLCV). *Nbpen3* plants displayed slightly enhanced symptoms (Fig. 6a), but no difference in viral CP accumulation was detected when compared to the WT plants (Fig. 6b). In spite of similar viral accumulation, TYLCV was found outside of the phloem in *NbPEN3* knock-out *N. benthamiana* plants (Fig. 6c), indicating that phloem restriction of geminiviruses is likely controlled by PEN3.

To further assess if PEN3 mediates phloem restriction to viruses in general, potato leaf roll virus (PLRV), an aphid-transmitted phloem-limited RNA virus, was used to inoculate WT and *nbpen3* mutant plants. *Nbpen3* plants were more susceptible to PLRV infection than the WT, displaying severe downward leaf curling and higher viral accumulation at 14 dpi (Fig. 6d, e). Strikingly, immunolocalization of the viral CP in cross-sections of leaf veins demonstrates that PLRV exits the phloem and reaches parenchyma cells in *nbpen3* mutant plants only (Fig. 6f). These results allow the conclusion that PEN3 also determines phloem limitation of an RNA virus.

## Discussion

Many plant viruses, such as poleroviruses, criniviruses and most begomoviruses, are phloem-restricted in natural infections, which correlates with their transmission by phloem-feeding insect vectors[19–21]. However, how viruses are restricted to the phloem and what the biological significance of this tissue tropism is, are long-standing questions in plant virology. The different tissue tropisms between TbCSV isolates provide a precious opportunity to investigate these questions. TbCSV (Y35) is an example of such tissue limitation, and its invasion can be recognized by the plant immune system, leading to strong callose deposition surrounding the infected vasculature (Fig. 3f, Supplementary Fig. 5). This response presumably leads to a physical constriction of plasmodesmata associated with decreased trafficking through these channels, and therefore the constraint of lateral viral movement. TbCSV(Y41), however, is capable of compromising callose accumulation during infection through the action of an additional chloroplast-localized C4 effector, cC4. cC4 targets chloroplasts and interferes with SA signaling, promoting PEN3 degradation and compromising callose-mediated phloem restriction. Surprisingly, the difference in tissue tropism between TbCSV(Y35) and TbCSV(Y41) is not reflected in increased virus accumulation (Fig. 1c). This phenomenon might be

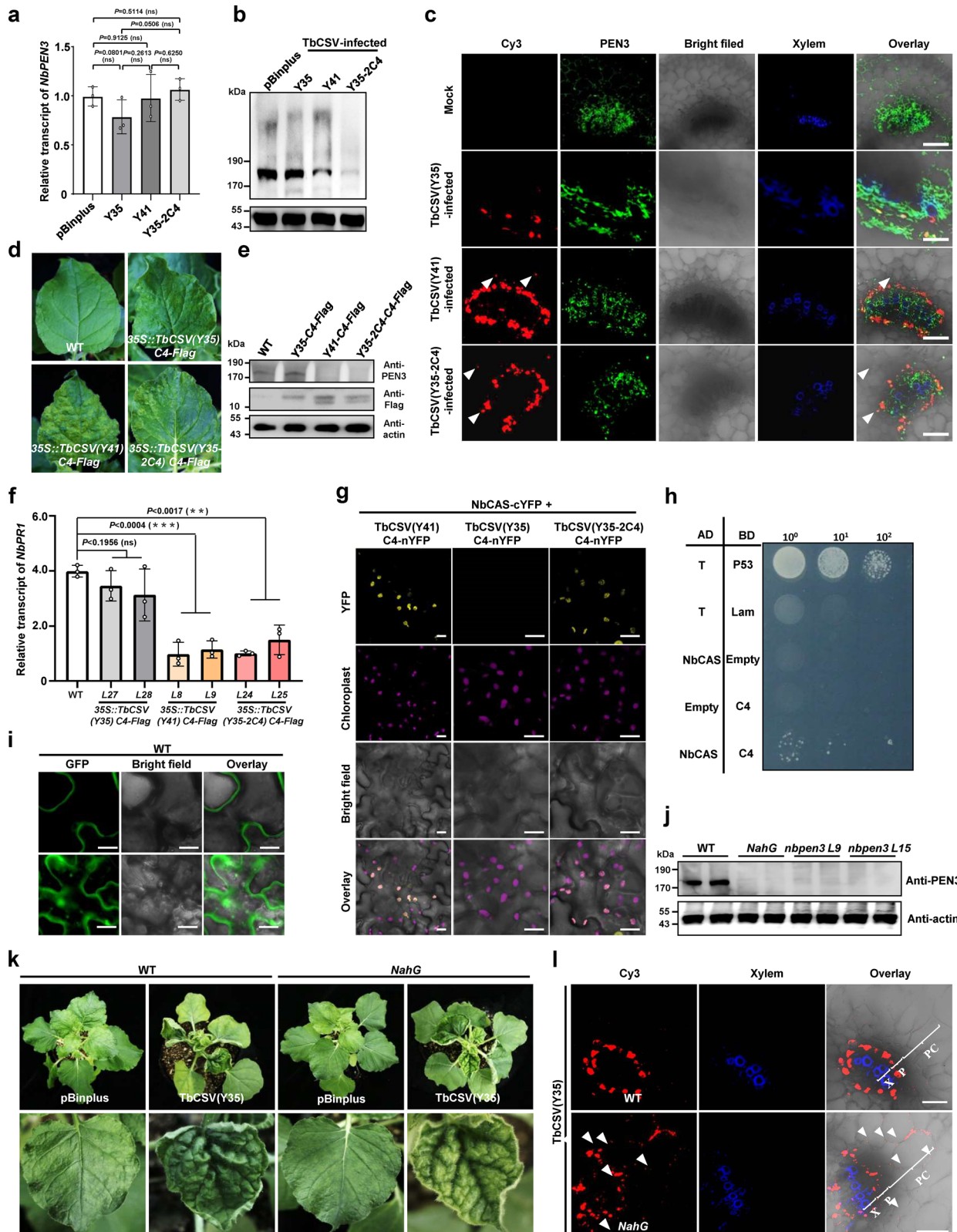

explained by the fact that geminivirus promoters can display phloem-specific activity, which confers higher replication efficiency in the phloem[22]; therefore, escaping this tissue would not result in a significant increase in viral load. Determining TbCSV replication efficiency in phloem companion vs parenchyma cells will shed light on this observation. Importantly, the finding that phloem restriction of the unrelated, aphid-transmitted RNA virus PLRV also depends on

PEN3 suggests a general role of this protein in mediating phloem limitation of viruses.

Chloroplast immunity is emerging as a cornerstone of plant defence[23-31]. In this study, we show that the chloroplast-localized C4 encoded by the Y41 allele of TbCSV interferes with SA signaling, likely through its direct interaction with NbCAS1 and/or by disrupting the redox balance in chloroplasts. Impaired SA signaling destabilizes PEN3,

**Fig. 4 | Geminiviruses compromise NbPEN3-mediated phloem restriction through encoding a chloroplast-targeted C4 protein. a** Relative transcript of *NbPEN3* in uninfected, TbCSV(Y35)-, TbCSV(Y41)- or TbCSV(Y35-2C4)-infected *N. benthamiana* plants. Relative accumulation of *NbPEN3* transcripts is normalized against that of *Actin*. Statistical differences were analyzed by a two-sided, unpaired Student's *t*-test (ns: not significant). Data are present as mean values ± SD of three biological replicates. Individual *P*-values are denoted above the comparison lines. **b** Western blot of NbPEN3 in uninfected, TbCSV(Y35)-, TbCSV(Y41)-, or TbCSV(Y35-2C4)-infected *N. benthamiana* plants. NbPEN3 was detected using a commercial antibody. Accumulation of Actin is shown as a control. **c** Immunofluorescence detection of TbCSV and NbPEN3 in sections *N. benthamiana* plants infected by TbCSV(Y35), TbCSV(Y41) or TbCSV(Y35-2C4), or mock-inoculated. The distribution of TbCSV(Y35), TbCSV(Y41) or TbCSV(Y35-2C4) was visualized in a cross-section of leaf vein tissues using an antibody against the TbCSV coat protein (CP) (red). The distribution of NbPEN3 was visualized in a cross-section using an antibody against PEN3 (green). Autofluorescence of highly lignified tissues is shown in blue. White arrowheads indicate TbCSV CP in parenchyma cells. Scale bar = 50 μm. **d** Phenotype of *35S::TbCSV(Y35) C4-Flag*, *35S::TbCSV(Y41) C4-Flag*, and *35S::TbCSV(Y35-2C4) C4-Flag* transgenic *N. benthamiana* plants. **e** Western blot of NbPEN3 in wild-type (WT), *35S::TbCSV(Y35) C4-Flag*, *35S::TbCSV(Y41) C4-Flag*, and *35S::TbCSV(Y35-2C4) C4-Flag* transgenic *N. benthamiana* plants. As in (**b**). **f** Relative transcript accumulation of *NbPR1* in WT, *35S::TbCSV(Y35) C4-Flag*, *35S::TbCSV(Y41) C4-Flag*, and *35S::TbCSV(Y35-2C4) C4-Flag* transgenic *N. benthamiana* plants. Relative accumulation of *NbPR1* transcripts is normalized to that of the *Actin* transcript. Statistical differences were analyzed by two-sided, unpaired Student's *t*-test (ns: not significant, **P < 0.01, ***P < 0.001). Data are present as mean values ± SD of three biological replicates. Individual *P* values are denoted above the comparison lines. **g, h** Identification of the interaction between NbCAS and the chloroplast-localized C4 in vivo and in yeast. **g** BiFC analysis of the interaction between the chloroplast-localized C4 and NbCAS in epidermal cells of H2B-RFP transgenic *N. benthamiana* plants. The combination of NbCAS-cYFP/TbCSV(Y35) C4-nYFP serves as a negative control. Scale bar = 20 μm. **h** Identification of the interaction between NbCAS and TbCSV(Y41) C4 in yeast. The yeast strain Gold co-transformed with the indicated plasmids was subjected to 10-fold serial dilutions and grown on a SD/-Leu/-Trp/-His medium. **i** Subcellular localization of GFP-NbPEN3 in epidermal cells of WT or *NahG* transgenic *N. benthamiana* plants. Scale bar = 20 μm. **j** Western blot analysis of NbPEN3 in WT and *NahG* transgenic *N. benthamiana* plants. **k** Viral symptoms induced by TbCSV(Y35) in WT and *NahG* transgenic *N. benthamiana* plants at 14 dpi. **l** Immunofluorescence detection of TbCSV(Y35) in sections of WT and *NahG* transgenic *N. benthamiana* plants. The distribution of TbCSV(Y35) was visualized in a cross-section of leaf vein tissues using an antibody against the TbCSV coat protein (CP) (red). Autofluorescence of highly lignified tissues is shown in blue. PC, parenchyma; P, phloem; X, xylem. White arrowheads indicate TbCSV in parenchyma cells. Scale bar = 50 μm. In TbCSV(Y35)-infected *NahG* transgenic *N. benthamiana* plants, over 60% of images show virus signal in parenchyma cells. At least five individual samples of TbCSV(Y35)-infected *NahG* transgenic *N. benthamiana* plants from three independent biological replicates were used for immunochemistry and statistical analysis. Experiments in (**c, i, j, l**) were repeated at least three times with similar results.

which in turn compromises virus-induced callose deposition and ultimately virus phloem restriction. This model is supported by several pieces of evidence: (1) The expression of the SA-responsive gene *PR1* is repressed in both *35S::TbCSV(Y41) cC4* and *35S::TbCSV(Y35-2C4) cC4* transgenic *N. benthamiana* plants, indicating a diminished SA response, but not in *35S:TbCSV(Y35)* (Fig. 4f); (2) PEN3 stability is regulated in a SA-dependent manner (Fig. 4j, Supplementary Fig. 9); (3) TbCSV(Y35) escapes the phloem and reaches parenchyma cells in the *nbpen3* mutant, which correlates with the lack of observable callose deposition (Fig. 3e, f, Supplementary Fig. 5). These results suggest a scenario in which TbCSV(Y41) compromises the PEN3-mediated viral phloem restriction through the action of a chloroplast-localized C4.

C4 is the most divergent canonical geminiviral protein, and the C4 proteins encoded by different geminiviruses have different functions and properties. Even though C4 from TYCLV could shuttle from the plasma membrane to chloroplasts during infection to dampen SA signaling[8], its efficiency might not suffice to ensure phloem escape. Further efforts to conduct a comparative study between the strengths of these viral proteins as SA suppressors and their ability to mediate phloem escape will shed light on this apparent conundrum.

The question remains of what the tell-tale of the viral infection is that triggers activation of defence and, ultimately, PEN3-mediated callose deposition. Viruses are devoid of canonical pathogen-associated molecular patterns (PAMPs), inducers of plant immunity; however, in geminiviruses, the replication-associated protein (Rep) or its activity has been shown to activate defense responses[8,32], and the genomic DNA has also been proposed to be perceived[33–35]. Viral RNA is believed to act as PAMP[36,37]. Nevertheless, the nature and mechanism of viral perception in the phloem await clarification.

Phloem restriction has been generally regarded as the result of the deployment of plant defence responses that the invading virus is unable to suppress, although the exact underlying molecular mechanisms have so far remained elusive. Our results show that phloem restriction of the geminivirus TbCSV and other viruses likely relies on PEN3-dependent virus-induced callose deposition, which probably results from recognition of the viral invasion. The chloroplast-localized viral C4 protein encoded by the TbCSV Y41 C4 allele, however, can suppress SA signaling, leading to a decreased accumulation of PEN3, impaired callose deposition, and phloem escape of the virus. This enlarged tissue tropism, nevertheless, has a

toll in insect transmission, since a lower proportion of the viral population is accessible to the phloem-feeding insect vector (Fig. 7). The decreased transmission efficiency imposed by phloem escape may underpin the observation that TbCSV strains carrying the Y41 allele are not dominant in nature, despite exhibiting an expansion in host range, and, more generally, that numerous viruses transmitted by phloem-feeding insects remain phloem-restricted through evolution.

## Methods
### Plasmids construction
The *NbPEN3* coding sequence (CDS) was obtained from cDNA derived from *N. benthamiana* plants and cloned into the pGD-GFP vector for the observation of NbPEN3 subcellular localization; *TbCSV(Y35) C4*, *TbCSV(Y41) C4*, and *TbCSV(Y35-2C4) C4* were cloned into the pCambia-Flag vector individually for the generation of transgenic *N. benthamiana* plants. For yeast two-hybrid assays (Y2H) and bimolecular fluorescence complementation assays (BiFC), CDS of *NbCAS1*, *TbCSV(Y35) C4*, *TbCSV(Y41) C4*, *TbCSV(Y35-2C4) C4* were recombined into pGADT7, pGBKT7, p2YN, and p2YC, respectively, generating AD-NbCAS1, BD-TbCSV(Y41) C4, TbCSV(Y35) C4–nYFP, TbCSV(Y41) C4–nYFP, TbCSV(Y35-2C4) C4–nYFP, and NbCAS1-cYFP. All cDNAs were PCR-amplified using the KOD-Plus-Neo High-Fidelity DNA polymerase (TOYOBO). The resulting PCR fragments were cloned individually into the destination vectors by using the One-Step Cloning Kit (Vazyme). All primers used for plasmid construction are listed in Supplementary Table 3.

### Plant materials, growth conditions, and generation of transgenic plants
Wild-type (WT) and transgenic *NahG* or histone 2B (H2B)-RFP *N. benthamiana* plants were cultivated in a greenhouse at 26 °C under a 16 h light/8 h dark photoperiod, and seedlings at an approximately 5-leaf stage were used for the experiments. The tomato (*Solanum lycopersicum*) cultivar 'Ailsa Craig' was used in this study. Seeds were sown in trays filled with a blend of peat and vermiculite in a 2:1 ratio (v:v). Upon full development of the second true leaf, seedlings were transplanted into pots containing the same peat and vermiculite mixture. Seedlings were supplied with 1/2 Hoagland's nutrient solution every two days and were cultivated in a greenhouse equipped with LED lighting, maintaining daytime temperatures around 23–28 °C and nighttime

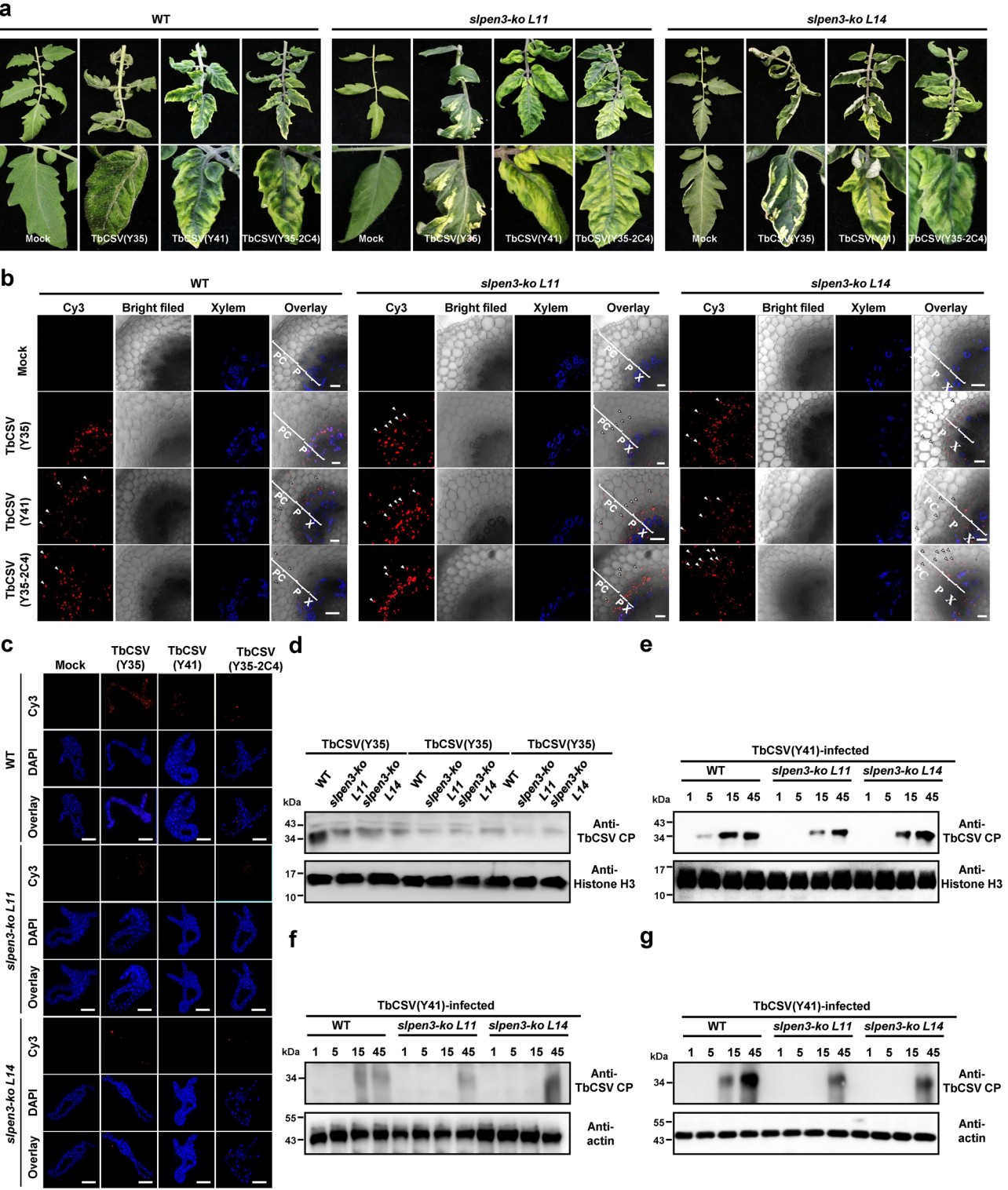

temperatures around 18–23 °C, and seedlings at an approximately 3-leaf stage were used for the experiments. CRISPR/Cas9-based knockout of *NbPEN3* and *SlPEN3* in *N. benthamiana* and tomato plants was generated by transformation with the binary vector pBGK01 (Biogle, Changzhou, China) in fusion with a single-guide RNA designed to target the ORFs. The synthesized sgRNA oligo was annealed and then ligated to pBGK01 with One-Step Cloning Kit (Vazyme, Nanjing, China). T2 homozygous transgenic plants were used for subsequent experiments. For overexpression of TbCSV(Y35) C4, TbCSV(Y41) C4 and TbCSV(Y35-2C4) C4, the plasmids carrying *35S::TbCSV(Y35) C4-*

*Flag*, *35S::TbCSV(Y41) C4-Flag* and *35S::TbCSV(Y35-2C4) C4-Flag* were used for transformation in *N. benthamiana* plants. Immunoblot was conducted to verify the gene expression.

### Immunocytochemistry for protein localization in sections of plant tissues

Immunocytochemichemistry for protein localization of plant tissues was performed essentially as described previously[38]. Leaf tissues (5 mm × 10 mm) of *N. benthamiana* or *S. lycopersicum* plants infected by different TbCSV isolates were fixed in 4% paraformaldehyde at 4 °C

**Fig. 5 | Phloem-limitation enabled by PEN3 favors acquisition of the virus by the insect vector. a** Viral symptom induced by TbCSV(Y35), TbCSV(Y41) and TbCSV(Y35-2C4) in wild-type (WT) and *slpen3* knock-out tomato plants at 30 days post-inoculation (dpi). Two independent *slpen3* knock-out tomato plants and yellowing symptoms in the leaf lamina of WT and *slpen3* tomato plants are shown. Photographs were taken at 30 dpi. **b** Immunofluorescence detection of TbCSV in sections of WT and *slpen3* knock-out tomato plants infected by TbCSV(Y35), TbCSV(Y41), or TbCSV(Y35-2C4). The distribution of TbCSV(Y35), TbCSV(Y41), or TbCSV(Y35-2C4) was visualized in a cross-section of leaf vein tissues using an antibody against the TbCSV coat protein (CP) (red). Autofluorescence of highly lignified tissues is shown in blue. PC, parenchyma; P, phloem; X, xylem. White arrowheads indicate TbCSV in parenchyma cells. Scale bar = 50 μm. **c** Immunofluorescence detection of TbCSV in midguts of Zhejiang II whiteflies after feeding on WT or *slpen3* knock-out tomato plants infected by TbCSV(Y35), TbCSV(Y41) or TbCSV(Y35-2C4). TbCSV was detected using a monoclonal antibody against the TbCSV coat protein (CP), followed by a commercial 549-conjugated secondary antibody (red), and nuclei were stained with DAPI (blue). At least 10 midguts were examined for each treatment. Scale bar = 100 μm. Representative images are shown. **d** Immunoblot analysis of virus accumulation in whiteflies after feeding on TbCSV(Y35)-, TbCSV(Y41)-, or TbCSV(Y35-2C4)-infected WT and *slpen3* knock-out tomato plants. Total protein of 150 whiteflies was extracted for immunoblot per treatment. Histone H3 was used as a loading control. **e–g** Immunoblot analysis of TbCSV(Y35) (**e**), TbCSV(Y41) (**f**), and TbCSV(Y35-2C4) (**g**) accumulation in WT tomato plants inoculated with 1, 5, 15, or 45 viruliferous Zhejiang II whiteflies acquiring virus from TbCSV(Y35)-, TbCSV(Y41)-, or TbCSV(Y35-2C4)-infected WT or *slpen3* knock-out tomato plants. TbCSV was detected using a monoclonal antibody against the TbCSV coat protein (CP). Actin was used as a loading control. Tomato plants were given an inoculation access period of 48 h with 1, 5, 15, or 45 viruliferous whiteflies using clip cages in three independent biological replicates. Total protein was extracted from WT tomato plants at 14 days after the inoculation access period. Experiments in (**b–d**) were repeated at least three times with similar results.

overnight. The fixative solution was removed, and the materials were washed three times for 5-10 min each with PBST at room temperature. Then the samples were dehydrated sequentially in 25%, 50%, 75%, and 96% ethanol for 60 min each at room temperature. After dehydration, the samples were incubated with wax/ethanol (1:2), wax/ethanol (1:1), wax/ethanol (2:1) and pure wax for 60 min each at 37 °C. Materials were then transferred into a Petri dish, placed evenly at the bottom of the dish, arranged in a final position, and pure wax was poured onto them. The material was placed at 4 °C for 20 min, to let the wax solidify. The material was then sectioned with a thickness of 6–8 μm by using a razor blade and transferred onto a microscope slide for 2 h. The slides were incubated sequentially in 99%, 90%, and 50% ethanol and PBS for 15 min each at room temperature for dewaxing and rehydration. Then the sections were blocked with 2% BSA for 2 h at room temperature, followed by incubation with anti-TbCSV CP monoclonal antibody (1:400) or Anti-PEN3 polyclonal antibody (PHYTOAB, Catalog: PHY0177A) (1:200) in PBST containing 1% BSA overnight at 4 °C and four washes with PBST for 10 min each at room temperature. Next, sections were incubated with 1% BSA solution with goat anti-mouse (1:500) 549-conjugated secondary antibody (Earthox, Catalog: E032410-01) or goat anti-rabbit (1:500) 488-conjugated secondary antibody (Earthox, Catalog: E032220-01) for 2 h at room temperature. After secondary antibody incubation, the material was washed with PBST three times for 10 min each, then covered with a cover glass and observed using a Zeiss LSM780 confocal microscope. The immunofluorescence intensity of TbCSV signal in the phloem of leaf vein sections (fluorescence intensity/selected area) in different treatments was measured using ImageJ software; at least 25 areas from three independent biological replicates were selected in each condition.

## Immunofluorescence assay

Immunofluorescence detection of viruses in Zhejiang II whitefly midguts and primary salivary glands was performed essentially as described previously[39,40]. Intact midguts of whiteflies and intact primary salivary glands of whiteflies at required developmental stages were dissected after a 48 h-access acquisition period (AAP) on virus-infected tomato plants, and the specimens were incubated and fixed in 4% paraformaldehyde for 1 h at room temperature. Samples were then washed in PBST containing 0.5% Triton X-100 three times for 30 min each. The specimens were blocked with 1% BSA for 2 h at room temperature, followed by incubation with anti-PLRV CP polyclonal antibody (1:100) or anti-TbCSV CP monoclonal antibody (1:400) in PBST containing 1% BSA overnight at 4 °C and washed with PBST containing 0.5% Triton X-100 three times for 30 min each. Samples were incubated with 1% BSA solution with goat anti-mouse (1:500) 549-conjugated secondary antibody (Earthox, Catalog: E032410-01) for 2 h at room temperature. All specimens were washed with PBST three times for 30 min each, and DNA was stained with 100 mM 4, 6-diamidno-2-phenylindole (DAPI) for 30 min at room temperature. For each experiment, 10 midguts or primary salivary glands were examined by using a Zeiss LSM780 confocal microscope.

## Electron microscopy

Plant tissues (1 mm × 4 mm) were excised from leaves of WT, *35S::TbCSV(Y35) C4-Flag*, *35S::TbCSV(Y41) C4-Flag* or *TbCSV(Y35-2C4) C4-Flag* transgenic *N. benthamiana* plants. The sampled tissues were fixed in 2.5% glutaraldehyde and 1% osmium tetroxide (both in 100 mM phosphate buffer, pH 7.0). The fixed tissues were embedded in Epon 812 resin as instructed by the manufacturer (SPI-EM, Division of Structure Probe, Inc., West Chester, USA). Ultrathin sections (70 nm) were cut from embedded tissues using the Ultracut E Ultramicrotome (Reichart-Jung, Vienna, Austria) and mounted on formvar-coated grids. Thin sections were then stained with uranyl acetate for 10 min followed by lead citrate for 10 min. The stained sections were examined under an electron microscope (HT7820; Hitachi, Japan).

## Detection of virus transmission efficiency

Three-leaf stage tomato plants were inoculated with the infectious clones of TbCSV(Y35), TbCSV(Y41), or TbCSV(Y35-2C4). The whitefly adults were placed on TbCSV(Y35)-, TbCSV(Y41)-, or TbCSV(Y35-2C4)-infected tomato plants to feed for a period of 48 h to acquire the virus. These adults were then collected individually, and a given number of adults (1, 5, or 15) were placed on new individual test tomato plants at 21 days post-inoculation (dpi) to feed for another period of 72 h for virus transmission by using clip cages in three independent biological replicates. Non-viruliferous whiteflies were used as a mock treatment. Immediately after the 72 h-virus transmission period, the whitefly adults were removed, and the test plants were sprayed with imidacloprid (20 mg/L) to kill all the whitefly eggs produced during the 72 h inoculation period. The test plants were further cultured for 21 days, and the virus infection status of the test plants was then determined by Immunoblotting.

## Callose detection and quantification

Callose staining was performed essentially as described previously[41]. Callose of wild-type (WT) and *nbpen3-ko* transgenic *N. benthamiana* plants was stained with 0.05 mg/mL aniline blue for 20 min at room temperature. The sections were washed with water three times, then the callose was detected at 405 nm wavelength by using a Zeiss LSM780 laser scanning microscope. The relative intensity of the fluorescent signal in the phloem (fluorescence intensity/selected area) in different treatments was measured by using ImageJ software; at least 10 areas from three independent biological replicates were selected in each treatment. Callose quantification was performed by measuring the callose deposition through dot-blot and antigen coating plate (ACP) ELIZA kit (Bioroyee, Catalog: RE5372). For dot-blot assays,

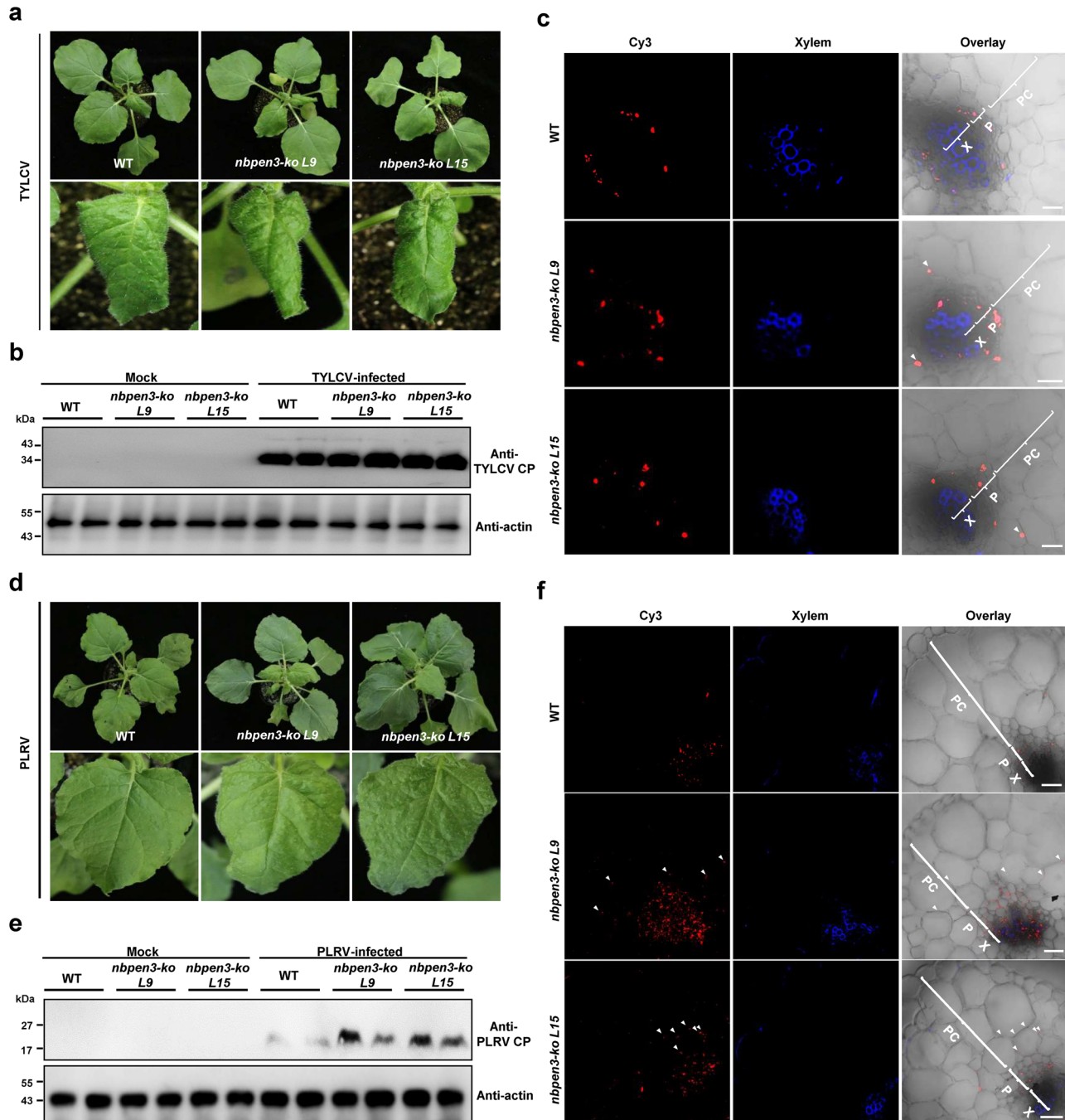

**Fig. 6 | PEN3 determines phloem limitation of other viruses. a** Symptoms of TYLCV-infected WT and *nbpen3 N. benthamiana* plants at 14 days post-inoculation (dpi). Upper panel shows the phenotype of leaves in WT and *nbpen3 N. benthamiana* plants. Lower panel shows the downward leaf curling induced by TYLCV. **b** Immunoblot analysis of TYLCV accumulation in WT and *nbpen3 N. benthamiana* plants at 14 dpi. Actin was used as loading control. **c** Immunofluorescence detection of TYLCV in cross-sections of leaf veins from infected WT and *nbpen3 N. benthamiana* plants. The distribution of TYLCV was visualized using antibodies against TYLCV CP (red). Autofluorescence of highly lignified tissues is shown in blue. PC, parenchyma; P, phloem; X, xylem. White arrowheads indicate TYLCV in

parenchyma cells. Scale bar = 50 µm. **d** Symptoms induced by PLRV in WT and *nbpen3 N. benthamiana* plants at 14 days post-inoculation (dpi). Lower panel shows the downward leaf curling induced by PLRV. **e** Immunoblot analysis of PLRV accumulation in WT and *nbpen3 N. benthamiana* plants at 14 dpi. Actin was used as loading control. **f** Immunofluorescence detection of PLRV in cross-sections of leaf veins from infected WT and *nbpen3 N. benthamiana* plants. The distribution of PLRV was visualized using antibodies against PLRV CP (red). Autofluorescence of highly lignified tissues is shown in blue. PC, parenchyma; P, phloem; X, xylem. White arrowheads indicate PLRV in parenchyma cells. Scale bar = 50 µm. Experiments in (**b**, **c**, **e**, **f**) were repeated three times with similar results.

0.2 g of leaf tissues of WT and transgenic *N. benthamiana* plants were homogenized in 200 µl of isolation buffer (50 mM Tris-HCl, pH 8.0, 150 mM NaCl, 5 mM DTT, 1 mM MgCl$_2$). The homogenate was filtered through a 47 µm nylon mesh, the flow-through was centrifuged at 8000 g for 10 min, and the supernatant was spotted onto

nitrocellulose membranes and allowed to dry. The membrane harboring proteins was incubated with 3% BSA in TBST buffer for 1 h at room temperature. The membrane was then incubated with TBST containing HRP-conjugated anti-callose antibody or anti-actin antibody at 4 °C overnight. Next, the membrane was washed with TBST

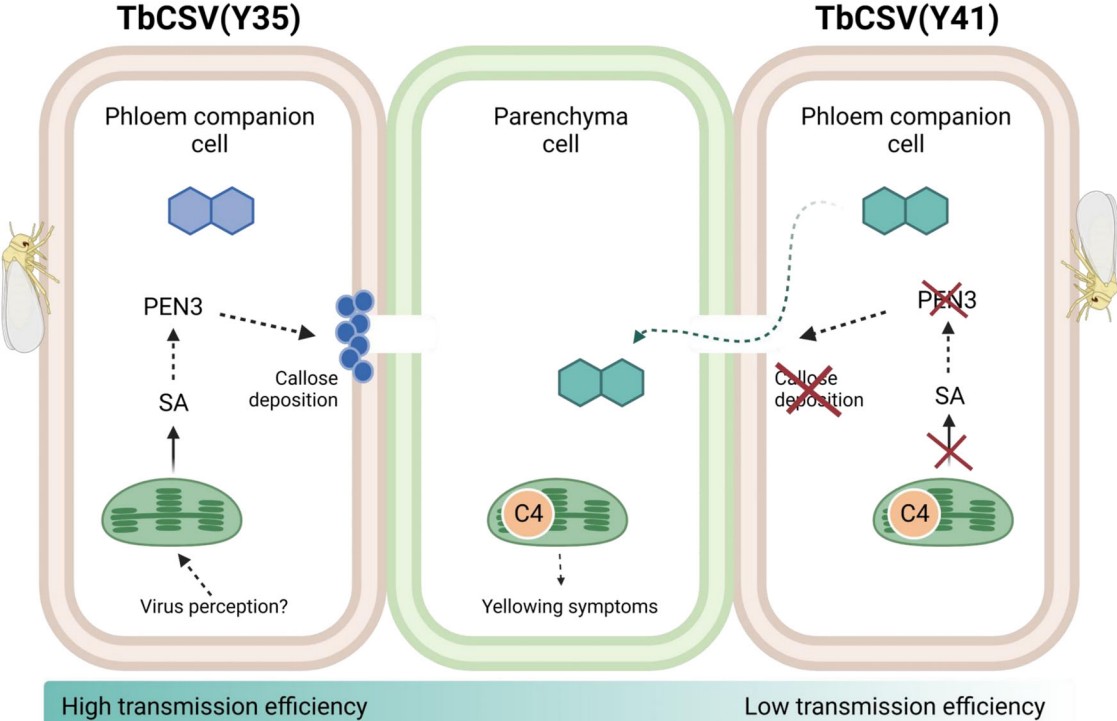

**Fig. 7 | Proposed mechanism underlying phloem restriction in geminiviruses and its escape by TbCSV(Y41).** Virus-induced PEN3-mediated callose deposition is critical for phloem limitation of geminiviruses, which favors acquisition and subsequent transmission by the phloem-feeding insect vector, the whitefly *B. tabaci*. In TbCSV(Y41), the virus-encoded chloroplast-localized C4 protein interferes with SA signaling and inhibits PEN3 accumulation, compromising phloem restriction; this expansion in tissue tropism, however, has a toll on the acquisition by the insect vector, since a lower proportion of the viral population is accessible, ultimately decreasing the virus' epidemic rate.

three times, and callose bound on the membrane was detected with the ECL western blotting kit (4 A Biotech). For ACP ELIZA, 0.2 g of leaf tissues of WT and transgenic *N. benthamiana* plants were homogenized 200 μl of isolation buffer (50 mM Tris-HCl, pH 8.0, 150 mM NaCl, 5 mM DTT, 1 mM $MgCl_2$). The homogenate was filtered through a 47 μm nylon mesh, the flow-through was centrifuged at 8000 g for 10 min, and the resulting supernatant was incubated in a 96-well microtiter plate at 4 °C overnight, respectively. The plate coated with callose-contained supernatant was washed with PBST buffer three times. Then, PBST containing HRP-conjugated anti-callose antibody was added to the wells coated with callose at 37 °C for 2 h. After washing four times with PBST, an ACP ELIZA was performed to detect the callose bound on the plate.

### Quantitative RT-PCR
Total RNA was extracted from *N. benthamiana* leaf tissues using TRIzol reagent (Invitrogen) according to the manufacturer's instructions. First-strand synthesis was performed using ReverTra Ace qPCR RT Master Mix with gDNA Remover (TOYOBO). Real-time PCR was conducted using SYBR Green I Master (Vazyme) according to the manufacturer's instructions. Primers were used for the amplification of the targets, and the efficiencies of all primers were verified by normal RT-PCR, gel electrophoresis, and melting curve analysis. Primer sequences are listed in Supplementary Table 3. Expression of the *Actin* gene was used as an internal control. The value of each bar indicates the average value of three independent measurements.

### Semi-in vivo protein degradation assay
Semi-in vivo protein degradation assays were performed essentially as described previously[42,43]. For semi-in vivo protein degradation analysis,

a GFP-NbPEN3 sample was harvested from the leaves infiltrated with Agrobacterium cultures expressing GFP-NbPEN3 in wild-type or *NahG* transgenic *N. benthamiana* plants treated with ddH₂O or 50 μM SA for 12 h. The samples were extracted gently with native extraction buffer (50 mM Tris-HCl, pH 8.0, 0.5 M sucrose, 1 mM $MgCl_2$, 10 mM EDTA, 5 mM dithionthreitol). For analysis of the GFP-NbPEN3 degradation, the plant extract harboring GFP-NbPEN3 was mixed with cyclohex-imide (CHX; 100 mM), then ATP was added (to a final concentration of 20 mM). These samples were agitated in an Eppendorf Thermomixer at 25 °C, and equal volumes of sthe ample were removed from the tube at different time points.

### Bimolecular fluorescence complementation (BiFC) assay
BiFC assays were performed on 5-week-old *N. benthamiana* leaves infiltrated with the combination of *Agrobacterium tumefaciens* EHA105 harboring p2YN-TbCSV(Y35) C4, producing TbCSV(Y35) C4−nYFP, or p2YN-TbCSV(Y41) C4 producing TbCSV(Y41) C4−nYFP, or p2YN-TbCSV(Y35-2C4) C4 producing TbCSV(Y35-2C4) C4−nYFP and *A. tumefaciens* EHA105 carrying p2YC-NbCAS1 producing NbCAS1-cYFP. The combination of NbCAS-cYFP/TbCSV(Y35) C4-nYFP serves as a negative control. Emission of the YFP interaction signal was detected using Zeiss LSM780 laser scanning microscope (Carl-Zeiss) at 48 h post-inoculation (hpi).

### Statistical analysis
All statistical analyses were done with the GraphPad Prism 8.0 software. A two-sided, unpaired Student's *t*-test was performed when appropriate. Details about the statistical approach used can be found in the figure legends. Data are represented as mean ± SD as indicated.

**Reporting summary**

Further information on research design is available in the Nature Portfolio Reporting Summary linked to this article.

## Data availability

*PEN3* sequences can be found in the Sol Genomics Network under accession numbers Niben101Scf04046g00002.1 (*NbPEN3*) and Solyc03g120980.3.1 (*SlPEN3*). The authors declare that the data supporting the findings of this study are available within the paper and its supplementary files. A reporting summary for this paper is available as a supplementary file. Source data are provided with this paper.

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

## Acknowledgments

This work was supported by the National Natural Science Foundation of China (W2411024) and the National Key Research and Development Program of China (2022YFD1400804) to Xueping Zhou, and the Basic Research Center of Innovation Program of Chinese Academy of Agricultural Sciences (CAAS-BRC-CB-2025-02) to Fangfang Li. The authors give thanks to Dr. Li Xie in the Analysis Center of Agrobiology and Environment Sciences, Zhejiang University, for the help with transmission electron microscopy.

## Author contributions

X.Z. and Y.M. conceived the project and designed the experiments. Y.M., Y.W., and F.L. performed experiments and analyzed data; X.Z. supervised work; Y.M. and R.L. prepared the figures; Y.M., R.L., and X.Z. wrote the manuscript, with input from all authors.

## Competing interests

The authors declare no competing interests.
