## [Transparent Peer Review file · Nature Communications]

Defence-mediated phloem restriction of a plant virus facilitates insect transmission

Corresponding Author: Professor Xueping Zhou

Version 0:

Reviewer comments:

Reviewer #1

(Remarks to the Author)

This research aims to shed light on a longstanding puzzle in plant virology; why certain viruses stay limited to the phloem. The discovery that this phloem confinement actually boosts viral acquisition by insect vectors could have a significant impact on improving disease control approaches. The authors put a lot of effort into this study and have found some interesting results. However, there are still issues with the manuscript. Parts of the paper are hard to follow. Writing sometimes lacks clarity. The Methods section, in particular, is confusing and missing key details that are essential for reproducibility. In some cases, results are presented without explaining how they were actually obtained. There are also findings that seem to need further validation, but instead they're used to make strong conclusions without enough caution. Overall, the study is important, but need additional improvement.

Major comments:

1. Adding additional proteins to viral genome can make many unforeseen and unknown changes. The authors have generated a 35S::TbCSV(Y41) C4 and 35S::TbCSV(Y35-2C4) transgenic plants. It is critical to test callose deposition and transmission with these transgenics
2. The work does not provide proof that it is the callose that limit the virus to the phloem. To show a direct link, they need to control the callose levels with a callose synthase or beta-glucanase.
3. For the fluorescent images- you need to quantify the results. Especially the callose and virus levels, but also in how many pictures they saw changes
4. Need to verify there is less virus in the phloem. Even if the virus escape it does not mean virus levels are lower in the phloem.
5. Figure 2e- XCP2::GFP is a xylem marker, but authors present is a phloem marker. This is a serious mistake. Also, in 2E I see virus outside the phloem in both versions of the virus, and I don't see a difference between them. This need to be quantified also in tomato.

Minor comments:

1. Methods section needs major revision

The Methods section is difficult to follow and lacks the clarity and detail necessary for reproducibility. Several key experimental procedures are either missing or insufficiently explained:

- Missing or incomplete method descriptions:

- o No methods and controls are provided for certain key experiments, such as BiFC interaction assays.
- o Transgenic plants are used throughout the study, but there is no explanation of how they were generated, maintained, or selected.
- o Although plasmid construction is described, the rationale behind making these constructs is unclear, and it's not stated whether they were used for agroinfiltration or for generating stable transgenic lines.

-Disorganized structure:

- o The current structure makes it difficult to connect methods to specific results. Important details like the infection process, number of biological replicates, and plant material are either missing or scattered.

-Specific method issues:

o Callose quantification: This section lacks detail; what tissue was analyzed? Dot blot and ELISA are semi-quantitative methods that may be influenced by loading variability. These findings should be verified with confocal microscopy and quantification of callose deposits per unit area.

o Plasmodesmata permeability assay: CFDA diffusion can be affected by variables like leaf age, turgor pressure, and application technique. Measuring diffusion area is an indirect proxy for plasmodesmal function and should be interpreted with caution.

o Semi-in vivo protein degradation assay: The relevance of this assay to the main research question is unclear and should be justified.

-Statistical analysis: Statistical approaches are not clearly explained and should be more thoroughly described.

2. Lack of complementation and confirmation

• Many conclusions about the role of PEN3 in phloem restriction rely heavily on loss of function (NBpen3) mutants. However, no complementation assays are presented (e.g., reintroducing PEN3 into the mutant background), which would confirm the role of PEN3.

-The claim that chloroplast-localized C4 interferes with PEN3 function is contradicted by the TYLCV findings. TYLCV's C4 localizes to both the plasma membrane and chloroplasts, yet phloem escape is only observed in the NBpen3 mutant, not in wildtype, suggesting that PEN3 is indeed a barrier, but that the chloroplast-localized C4 alone may not be sufficient to disrupt it. This discrepancy needs to be addressed and interpreted more carefully.

Reviewer #2

(Remarks to the Author)

Reviewer #3

(Remarks to the Author)

The primary subject of this excellent paper is the Y41 allele of the tobacco curly shoot virus (TbCSV) C4 gene. Different TbCSVs differ in the C4 variants they encode. One allele (Y35) produces a protein targeted to the plasma membrane and confers limitation to the phloem. The Y41 allele produces two C4 protein variants, one targeted to the plasma membrane, the other to chloroplasts. The Y41 allele compromises phloem restriction to the extent that it reduces callose deposition and this permits escape of TbCSV9Y41 to parenchyma cells. Y41 also expands the virus host range significantly. Of particular interest, confining the virus to the phloem facilitates acquisition by the insect vector.

The data indicate that when both variants of C4 are present in tomato plants, the insects accumulate the virus to lower levels and transmission is lowered. Immunolocalization confirms that the Y41 allele compromises phloem restriction.

It is reasonable to speculate that the Y35 allele is more common because of better phloem transmission. The existence in nature of the Y41 allele, since it results in reduced phloem transmission, and is probably not as well transmitted from one plant to another, remains puzzling. It is probable that something is missing from the story, but this paper brings us substantially forward on the path to understanding the phenomenon and virus ecology in general.

I am not sure why, "Phloem restriction has been generally regarded as the result of the deployment of plant responses that the invading virus is unable to suppress..." (line 265). The virus benefits from not suppressing the plant's defenses which keep it from injuring the plant, its host. It benefits by restricting itself to the phloem from which it can be picked up by insects and carried to other plants. In that sense the Y41 variant, injuring the plant and reducing its phloem transmission, remains the true puzzle. Is it possible that Y41 is an ancestral variety, on a slow path to extinction?

In general, the data, with some exceptions, are valid and important to understanding the paper.

The paper is well written.

More specific suggestions:

Do not use the term PEN3 in the title unless it is explained in the title. It only puzzles the reader since they probably do not know it means.

Line 232. Perhaps "constraint of viral movement" should read "constraint of lateral viral movement."

Fig. 1: The legend refers to Yunnan, China, the province illustrated, but the fact that Yunnan is the entire province is not clear from the legend, and the name Yunnan is not included on the map.

Fig. 2d: This figure, central and critically important to the paper, is very good and very convincing. The authors may have missed something, however. Note that the virus is seen on both sides of the xylem. That is because the vascular bundles in the Solanaceae are bicollateral – with phloem on both sides. That should be mentioned in the paper and the figures should be labeled that way, for example 4I and others. See Ye, Z-H, *Annu Rev Plant Biol* 53: 183-202 (2002). Are migrant virus present on both sides?

Supplementary Fig. 4. The quantitative callose data are convincing (C, D). However, the microscopic analyses are less convincing. S4B is difficult to interpret. S4E is also difficult: entry of CFDA into the leaf would have been largely restricted by wax on the epidermis. Were the leaf surfaces abraded before CFDA was applied? Also, pictures should have been taken at increasing time intervals. I suggest omitting B, E and F – they are not necessary - unless more convincing arguments can be made.

Fig. 4f. This figure is important and convincing. However, is not clear why some of the bars are colored and other not. There seems to be no need.

Version 1:

Reviewer comments:

Reviewer #1

(Remarks to the Author)

Thank you for the additional experiments and explanations. There are still some important points to improve:

- A proper quantification of the fluorescent images is missing. Reporting approximate percentages is useful, but without knowing how many images were analyzed, how those percentages were calculated, or whether any statistical tests were applied, it's hard to assess the strength of the evidence. Since several of the paper's key conclusions rely on imaging data, particularly for virus movement and callose levels, it would really strengthen the study to include more standardized quantification, such as fluorescence intensity measurements or signal area, along with clear sample sizes and stats. If that's not feasible for all data, even a clearer explanation of how the current estimates were derived would help improve clarity and reproducibility.
- I don't understand what difference it makes for transmission if the viral titer is always the same. For the insect, the only thing that should matter is how much viral particles are in the phloem, isn't it?
- Figure 2e, can you verify its really the Sweet marker? I see mostly xylem being labelled. Can you compare with XCP2-GFP, the original marker?

Reviewer #2

(Remarks to the Author)

Reviewer #3

(Remarks to the Author)

Version 2:

Reviewer comments:

Reviewer #1

(Remarks to the Author)

Thank you for all the edits. I am sorry, maybe I things were not very clear. In the first review I asked to measure the viral titer in the phloem, and this wasn't added. Maybe I was not clear. I think this experiment is vital for this story connecting the phloem escape to the transmission. You can use the same method that you use for measuring phloem callose to thos there is actually less virus in the phloem after escaping.

REVIEWER COMMENTS

Reviewer #1 (Remarks to the Author):

This research aims to shed light on a longstanding puzzle in plant virology; why certain viruses stay limited to the phloem. The discovery that this phloem confinement actually boosts viral acquisition by insect vectors could have a significant impact on improving disease control approaches. The authors put a lot of effort into this study and have found some interesting results. However, there are still issues with the manuscript. Parts of the paper are hard to follow. Writing sometimes lacks clarity. The Methods section, in particular, is confusing and missing key details that are essential for reproducibility. In some cases, results are presented without explaining how they were actually obtained. There are also findings that seem to need further validation, but instead they're used to make strong conclusions without enough caution. Overall, the study is important, but need additional improvement.

Major comments:

1. Adding additional proteins to viral genome can make many unforeseen and unknown changes. The authors have generated a 35S::TbCSV(Y41) C4 and 35S::TbCSV(Y35-2C4) transgenic plants. It is critical to test callose deposition and transmission with these transgenics.

Our response: Following the reviewer's advice, we have tested callose accumulation in leaves of 35S::TbCSV(Y35) C4, 35S::TbCSV(Y41) C4 and 35S::TbCSV(Y35-2C4) transgenic *Nicotiana benthamiana* plants in dot blot assays (Supplementary Figure 7D). Unfortunately, *N. benthamiana* plants exhibit strong insect resistance, which makes testing transmission efficiency in these transgenic plants not feasible.

Nevertheless, for viral functional studies, we believe the best approach to evaluate biological relevance for infection is to measure the contribution of C4 in the context of the viral genome; our experiments with allele swaps clearly demonstrate that virus expressing chloroplast-localized C4 triggers the reduction in PEN3 content and callose deposition (Supplementary Figure 5 and Supplementary Figure 7).

2. The work does not provide proof that it is the callose that limit the virus to the phloem. To show a direct link, they need to control the callose levels with a callose synthase or beta-glucanase.

Our response: We thank the reviewer for this suggestion. It has been previously reported that callose is synthesized by 12 members of the glucan synthase-like (GSL) gene family in plants (Huang et al., 2009, *Journal of Plant Physiology*), which makes it extremely challenging to control callose levels through knocking out *GSLs* by using CRISPR-Cas9-mediated genome editing. In addition, *gsl* mutants can have deleterious effects on plant development (Wang et al., 2022, *Heliyon*). As an alternative approach, we treated WT and TbCSV-infected *N. benthamiana* plants with 2-DDG, an inhibitor of callose synthesis. Dot blot results showed that leaves treated with 2-DDG accumulate lower levels of callose

(Supplemental Figure 6A). Consistently, TbCSV(Y35) escapes the phloem to the surrounding parenchyma cells in 2-DDG-treated leaves (Supplemental Figure 6B, C). These results strongly support the idea that callose deposition plays a critical role in phloem restriction of virus.

3. For the fluorescent images– you need to quantify the results. Especially the callose and virus levels, but also in how many pictures they saw changes.

Our response: We thank the reviewer for their valuable suggestion. Callose levels were measured independently of imaging in Supplemental Figure 6A, Supplemental Figure 7D, and Supplemental Figure 10; the results of the quantification correlate with the differences observed in the images. TbCSV(Y35) is strictly limited to the phloem in *N. benthamiana* or *Solanum lycopersicum* (tomato) plants. However, TbCSV(Y41) and TbCSV(Y35-2C4) can be observed overcoming phloem restriction and reaching parenchyma cells in over 70% of the images in either host (Figure 2D, 3E and 5D). In TbCSV(Y35)-infected *pen3-ko N. benthamiana* or *S. lycopersicum* plants, about 30% of images show virus signal in parenchyma cells (Figure 3E and 5D). In TbCSV(Y35)-infected *N. benthamiana* plants treated with 2-DDG, the virus is detected in parenchyma cells in approximately 10% of images (Supplemental Figure 6B and 6C). While TYLCV and PLRV are tightly restricted to the phloem in WT *N. benthamiana* plants, over 30% of images show viral signal in parenchyma cells in infected *pen3-ko N. benthamiana* plants.

4. Need to verify there is less virus in the phloem. Even if the virus escape it does not mean virus levels are lower in the phloem.

Our response: Strikingly, the difference in tissue tropism between TbCSV isolates is not reflected in changes in virus accumulation (Figure 1C, Figure 2B and Figure 3D), suggesting the total viral titer remains unchanged and the accumulation of TbCSV(Y41) and TbCSV(Y35-2C4) in the phloem must therefore be lower. The reason for such a decrease of viral load in the phloem is discussed in the Discussion section (Lines 238-242; Lines 280-282).

5. Figure 2e– XCP2::GFP is a xylem marker, but authors present is a phloem marker. This is a serious mistake. Also, in 2E I see virus outside the phloem in both versions of the virus, and I don't see a difference between them. This need to be quantified also in tomato.

Our response: We apologize for this mistake and thank the reviewer for spotting it: the line used in these experiments is in fact SWEET11-GFP, and not XCP2-GFP; the labels have now been corrected. In Figure 2E, the viral signal (red) outside of the vasculature is indicated by arrowheads – both xylem (blue) and the bright field are shown for reference. The virus is detected in parenchyma cells in TbCSV(Y41) and TbCSV(Y35-2C4)-infected *SWEET11::GFP* transgenic *N. benthamiana* plants in more than 70% of images.

Minor comments:

1. Methods section needs major revision

The Methods section is difficult to follow and lacks the clarity and detail

necessary for reproducibility. Several key experimental procedures are either missing or insufficiently explained:

- Missing or incomplete method descriptions:

o No methods and controls are provided for certain key experiments, such as BiFC interaction assays.

Our response: Following the reviewer's advice, we have now modified the Methods section and added further experimental details for BiFC assays. In these experiments, the combination of NbCAS-cYFP/TbCSV(Y35) C4-nYFP serves as negative control.

o Transgenic plants are used throughout the study, but there is no explanation of how they were generated, maintained, or selected.

Our response: We thank the reviewer for spotting this omission. The description of plant materials, growth conditions, and generation, maintenance, and verification of transgenic lines have been added to the Methods section.

o Although plasmid construction is described, the rationale behind making these constructs is unclear, and it's not stated whether they were used for agroinfiltration or for generating stable transgenic lines.

Our response: According to the reviewer's advice, we have modified the description of plasmid construction and specified the rationale behind the generation of these constructs.

-Disorganized structure:

o The current structure makes it difficult to connect methods to specific results. Important details like the infection process, number of biological replicates, and plant material are either missing or scattered.

Our response: Following the reviewer's advice, we have now added details on the inoculation protocol, number of biological replicates, and plant materials in the Methods section as well as in figure legends.

-Specific method issues:

o Callose quantification: This section lacks detail; what tissue was analyzed? Dot blot and ELISA are semi-quantitative methods that may be influenced by loading variability. These findings should be verified with confocal microscopy and quantification of callose deposits per unit area.

Our response: We thank the reviewer for the valuable comments. We have added the experimental details of dot blot and ELISA in the Methods section; in both cases, actin was used as loading control (Supplemental Figure 6A, 7D and 10). Due to limitations in confocal imaging of phloem companion cells, it is difficult to quantification of callose deposits per unit area. We think the dot blot and ELISA could be credible ways to measure the callose accumulation.

o Plasmodesmata permeability assay: CFDA diffusion can be affected by variables like leaf age, turgor pressure, and application technique. Measuring diffusion area is an indirect proxy for plasmodesmal function and should be interpreted

with caution.

Our response: In the current version of the manuscript, we have omitted the CFDA diffusion results in Supplemental Figure 4E and 4F, also following the comments by Reviewer #3.

o Semi-in vivo protein degradation assay: The relevance of this assay to the main research question is unclear and should be justified.

Our response: To verify the role of SA in the regulation of PEN3 stability, semi-*in vivo* degradation assays were conducted. We have modified the text to clarify the relevance of semi-*in vivo* degradation assays to test the hypothesis that SA plays a critical role in PEN3 stability.

-Statistical analysis: Statistical approaches are not clearly explained and should be more thoroughly described.

Our response: We thank the reviewer for the valuable comment. We have added details on the statistical approaches in the Methods section.

2. Lack of complementation and confirmation

- Many conclusions about the role of PEN3 in phloem restriction rely heavily on loss of- function (NBpen3) mutants. However, no complementation assays are presented (e.g., reintroducing PEN3 into the mutant background), which would confirm the role of PEN3.

Our response: We drew the conclusions about the role of PEN3 in phloem restriction by using two independent *pen3-ko* mutant lines, an alternative to genetic complementation, and similar results were obtained. These results strongly support the notion that PEN3 indeed plays an important role in phloem restriction of viruses.

-The claim that chloroplast-localized C4 interferes with PEN3 function is contradicted by the TYLCV findings. TYLCV's C4 localizes to both the plasma membrane and chloroplasts, yet phloem escape is only observed in the NBpen3 mutant, not in wildtype, suggesting that PEN3 is indeed a barrier, but that the chloroplast-localized C4 alone may not be sufficient to disrupt it. This discrepancy needs to be addressed and interpreted more carefully.

Our response: C4 is the most divergent geminiviral protein; the C4 proteins encoded by different geminiviruses have different functions and properties. Even though C4 from TYCLV suppresses SA signalling, this might not be enough to ensure phloem escape. In order to make further conclusions, a comparative study between the strengths of these viral proteins as SA suppressors should be conducted in the future. Following the reviewer's advice, we have now included this point in the Discussion section.

Reviewer #2 (Remarks to the Author):

I co-reviewed this manuscript with one of the reviewers who provided the listed reports. This is part of the Nature Communications initiative to facilitate

training in peer review and to provide appropriate recognition for Early Career Researchers who co-review manuscripts.

Reviewer #3 (Remarks to the Author):

The primary subject of this excellent paper is the Y41 allele of the tobacco curly shoot virus (TbCSV) C4 gene. Different TbCSVs differ in the C4 variants they encode. One allele (Y35) produces a protein targeted to the plasma membrane and confers limitation to the phloem. The Y41 allele produces two C4 protein variants, one targeted to the plasma membrane, the other to chloroplasts. The Y41 allele compromises phloem restriction to the extent that it reduces callose deposition and this permits escape of TbCSV9Y41 to parenchyma cells. Y41 also expands the virus host range significantly. Of particular interest, confining the virus to the phloem facilitates acquisition by the insect vector.

The data indicate that when both variants of C4 are present in tomato plants, the insects accumulate the virus to lower levels and transmission is lowered. Immunolocalization confirms that the Y41 allele compromises phloem restriction.

It is reasonable to speculate that the Y35 allele is more common because of better phloem transmission. The existence in nature of the Y41 allele, since it results in reduced phloem transmission, and is probably not as well transmitted from one plant to another, remains puzzling. It is probable that something is missing from the story, but this paper brings us substantially forward on the path to understanding the phenomenon and virus ecology in general.

I am not sure why, “Phloem restriction has been generally regarded as the result of the deployment of plant responses that the invading virus is unable to suppress…” (line 265). The virus benefits from not suppressing the plant’s defenses which keep it from injuring the plant, its host. It benefits by restricting itself to the phloem from which it can be picked up by insects and carried to other plants. In that sense the Y41 variant, injuring the plant and reducing its phloem transmission, remains the true puzzle. Is it possible that Y41 is an ancestral variety, on a slow path to extinction?

Our response: We believe the fact that Y41 expands the host range might confer an advantage, due to its ability to multiply in alternative hosts, which may explain its existence in nature. It cannot be ruled out, however, that, as suggested by the reviewer, Y41 is either an ancestral variety, or on its way to being displaced by Y35.

In general, the data, with some exceptions, are valid and important to understanding the paper. The paper is well written.

More specific suggestions:

Do not use the term PEN3 in the title unless it is explained in the title. It only puzzles the reader since they probably do not know it means.

Our response: We thank the reviewer for this suggestion, and have modified the title accordingly.

Line 232. Perhaps “constraint of viral movement” should read “constraint of lateral viral movement.”

Our response: We thank the reviewer for this suggestion, and have modified the text as indicated.

Fig. 1: The legend refers to Yunnan, China, the province illustrated, but the fact that Yunnan is the entire province is not clear from the legend, and the name Yunnan is not included on the map.

Our response: We thank the reviewer for the valuable comments. We have modified the legends accordingly.

Fig. 2d: This figure, central and critically important to the paper, is very good and very convincing. The authors may have missed something, however. Note that the virus is seen on both sides of the xylem. That is because the vascular bundles in the Solanaceae are bicollateral - with phloem on both sides. That should be mentioned in the paper and the figures should be labeled that way, for example 4I and others. See Ye, Z-H, *Annu Rev Plant Biol* 53: 183-202 (2002). Are migrant virus present on both sides?

Our response: We thank the reviewer for pointing out this. In fact, the phloem-escaped virus could be detected on both sides of the vascular bundle (Figure 2E). This is now mentioned in the text (lines 110-112).

Supplementary Fig. 4. The quantitative callose data are convincing (C, D). However, the microscopic analyses are less convincing. S4B is difficult to interpret. S4E is also difficult: entry of CFDA into the leaf would have been largely restricted by wax on the epidermis. Were the leaf surfaces abraded before CFDA was applied? Also, pictures should have been taken at increasing time intervals. I suggest omitting B, E and F - they are not necessary - unless more convincing arguments can be made.

Our response: We thank the reviewer for the valuable comments. In this new version of the manuscript, we have removed former Supplementary Figure 4B, 4E and 4F in New Figure S4.

Fig. 4f. This figure is important and convincing. However, is not clear why some of the bars are colored and other not. There seems to be no need.

Our response: We have now modified Figure 4F, following the reviewer's advice.

REVIEWER COMMENTS

Reviewer #1 (Remarks to the Author):

Thank you for the additional experiments and explanations. There are still some important points to improve:

- A proper quantification of the fluorescent images is missing. Reporting approximate percentages is useful, but without knowing how many images were analyzed, how those percentages were calculated, or whether any statistical tests were applied, it's hard to assess the strength of the evidence. Since several of the paper's key conclusions rely on imaging data, particularly for virus movement and callose levels, it would really strengthen the study to include more standardized quantification, such as fluorescence intensity measurements or signal area, along with clear sample sizes and stats. If that's not feasible for all data, even a clearer explanation of how the current estimates were derived would help improve clarity and reproducibility.

Our response: We thank the reviewer for their valuable suggestion. Following the reviewer's advice, we quantified the mean intensity of phloem callose signal (fluorescence intensity/selected area) in different treatments measured by using ImageJ software, at least 10 areas were selected in each treatment for callose signal quantification (Figure S5A). Immunocytochemistry assays for protein localization in sections of plant tissues were performed by using at least five individual tissues each treatment. Statistics and details about samples number and imaging data were provided in figure legends and Methods and Materials Section.

- I don't understand what difference it makes for transmission if the viral titer is always the same. For the insect, the only thing that should matter is how much viral particles are in the phloem, isn't it?

Our response: Begomoviruses are vectored by phloem-feeding whitefly (Wang and Blanc, 2021, Annual Review of Entomology). Virus titer in the phloem determines the transmission efficiency of virus. Although the total viral titer remains unchanged, but the escape of TbCSV(Y41) and TbCSV(Y35-2C4) to the surrounding parenchyma leads to the lower accumulation of TbCSV(Y41) and TbCSV(Y35-2C4) in the phloem, which decreases the transmission efficiency of TbCSV(Y41) and TbCSV(Y35-2C4).

- Figure 2e, can you verify its really the Sweet marker? I see mostly xylem being labelled. Can you compare with XCP2-GFP, the original marker?

Our response: We thank the reviewer for their valuable suggestion. We provide the confocal micrograph of *XCP2::GFP* and *SWEET11::GFP* transgenic tissues in immunocytochemistry assays. GFP signal is coincided with autofluorescence of xylem in *XCP2::GFP* transgenic *Nicotiana benthamiana* plants. In *SWEET11::GFP* transgenic *Nicotiana benthamiana*

plants, GFP signal could be obviously detected in phloem of bicollateral bundle. And also, GFP was detected in some cells seem as xylem parenchyma cells. Although *AtSWEET11* is responsible for sucrose transportation in phloem, however, we have no reason to exclude other possibilities: (1) *AtSWEET* promoter might be active and functional in xylem parenchyma cells of *Nicotiana benthamiana* plants; (2) Cell types between xylem vessels are difficult to identify by using confocal microscope.

Reviewer #1 (Remarks to the Author):

Thank you for all the edits. I am sorry, maybe I things were not very clear. In the first review I asked to measure the viral titer in the phloem, and this wasn't added. Maybe I was not clear. I think this experiment is vital for this story connecting the phloem escape to the transmission. You can use the same method that you use for measuring phloem callose to thos there is actually less virus in the phloem after escaping.

Our response: We thank the reviewer for the valuable suggestion. Follow the reviewer's advice, we quantified the mean immunofluorescence intensity of TbCSV signal in the phloem of leaf vein sections (fluorescence intensity/selected area) in wild-type (WT) and *nbp3* knock-out *N. benthamiana* plants infected by TbCSV(Y35), TbCSV(Y41), or TbCSV(Y35-2C4). We find that phloem restriction of TbCSV(Y35) in wild-type plants was associated with higher virus accumulation in the phloem, while PEN3-mediated callose deposition triggered by TbCSV(Y35) infection can effectively restrict this viral strain and increase viral titer in the phloem (new Figure S5C). At least 25 areas from three independent biological replicates were selected for each experimental condition. Statistics and details about sample number are provided in the corresponding figure legend and in the Methods and Materials section.